# Synaptic mechanisms underlying modulation of locomotor-related motoneuron output by premotor cholinergic interneurons

**Filipe Nascimento[1], Matthew James Broadhead[1], Efstathia Tetringa[2], Eirini Tsape[2], Laskaro Zagoraiou[2], Gareth Brian Miles[1]\***

[1]School of Psychology and Neuroscience, University of St Andrews, St Andrews, United Kingdom; [2]Center of Basic Research, Biomedical Research Foundation of the Academy of Athens, Athens, Greece

**Abstract** Spinal motor networks are formed by diverse populations of interneurons that set the strength and rhythmicity of behaviors such as locomotion. A small cluster of cholinergic interneurons, expressing the transcription factor Pitx2, modulates the intensity of muscle activation via 'C-bouton' inputs to motoneurons. However, the synaptic mechanisms underlying this neuromodulation remain unclear. Here, we confirm in mice that Pitx2[+] interneurons are active during fictive locomotion and that their chemogenetic inhibition reduces the amplitude of motor output. Furthermore, after genetic ablation of cholinergic Pitx2[+] interneurons, M2 receptor-dependent regulation of the intensity of locomotor output is lost. Conversely, chemogenetic stimulation of Pitx2[+] interneurons leads to activation of M2 receptors on motoneurons, regulation of Kv2.1 channels and greater motoneuron output due to an increase in the inter-spike afterhyperpolarization and a reduction in spike half-width. Our findings elucidate synaptic mechanisms by which cholinergic spinal interneurons modulate the final common pathway for motor output.

**\*For correspondence:**
gbm4@st-andrews.ac.uk

**Competing interests:** The authors declare that no competing interests exist.

## Introduction

Locomotion is controlled by neuronal networks of the spinal cord, referred to as central pattern generators (CPGs), which can generate and sustain rhythmic patterns of muscle activity (*Arber, 2012*; *Gosgnach et al., 2017*; *Goulding, 2009*). The CPG for locomotion is located within the spinal cord and is formed by genetically distinct classes of interneurons that drive motoneuron activity in coordinated sequences that cause precise muscle activation necessary for behaviors such as walking or swimming. These spinal neurons are functionally diverse, as evidenced by the variety of different inhibitory, excitatory and modulatory transmitters they release to shape the pattern and frequency of motoneuron firing. Knowledge of the varying synaptic mechanisms by which interneuron subtypes participate in spinal circuits is crucial to better understand the flexibility and adaptability of spinal circuits in health and disease (*Goulding, 2009*; *Miles and Sillar, 2011*).

There are four broad, genetically distinct classes of spinal interneurons born in the ventral horn that are involved in motor function: V0, V1, V2 and V3 interneurons. V0 interneurons are derived from progenitors that express the developing brain homeobox protein 1 (Dbx-1) and can be further subdivided into V0$_V$ (ventral) or V0$_D$ (dorsal) interneurons based on the expression or absence, respectively, of the postmitotic determinant homeodomain protein even-skipped homeobox 1 (Evx1). V0 interneurons play an important role in left-right coordination (*Moran-Rivard et al., 2001*; *Lanuza et al., 2004*; *Pierani et al., 2001*). Less than 10% of the V0 neurons constitute a

subpopulation of interneurons identified by expression of the Paired-like homeodomain 2 (Pitx2) transcription factor. Pitx2[+] interneurons are preferentially clustered around the central canal and can be further subdivided into cholinergic (V0c) and glutamatergic (V0g) subtypes (*Zagoraiou et al., 2009*). V0c interneurons, receive excitatory inputs from spinal pathways involved in locomotion and innervate spinal motoneurons, forming 80–100 large synapses (*Zagoraiou et al., 2009*), termed C-boutons, on each motoneuron (*Li et al., 1995*). Recently, Rozani et al demonstrated that C-boutons of brainstem motoneurons also derive from Pitx2[+] interneurons (*Rozani et al., 2019*).

Unlike other V0-derived interneurons, V0c interneurons are not involved in left-right or extensor-flexor coordination. Instead, they seem to be important for modulating the amplitude of motor output. In wild-type mice, the activation of specific hindlimb muscles is increased when animals switch from walking to swimming. However, when cholinergic transmission at C-boutons is genetically perturbed, this task-dependent increase in muscle activation is significantly reduced (*Zagoraiou et al., 2009*). These findings support that V0c interneurons and their C-bouton inputs to motoneurons provide task-dependent modulation of motor output by enhancing motoneuron firing when greater muscle activation is required.

A range of postsynaptic proteins, including M2 muscarinic receptors, Kv2.1 channels and SK channels, are clustered at C-bouton synapses (*Wilson et al., 2004*; *Deardorff et al., 2014*). Previous work indicated that activation of M2 receptors in spinal cord slices increases motor output through M2 receptor-mediated inhibition of SK channels, which reduces the medium afterhyperpolarization (mAHP) allowing increased motoneuron firing rates (*Miles et al., 2007*). Recent work utilizing Kv2.1 channel blockers has demonstrated that Kv2.1 currents also regulate the repetitive firing of motoneurons (*Romer et al., 2019*). However, the exact mechanisms by which V0c-derived C-boutons modulate motor output remain unknown (*Deardorff et al., 2014*; *Witts et al., 2014*). Given the important role of V0c interneurons in motor control, and data implicating C-boutons as therapeutic targets for Amyotrophic Lateral Sclerosis and peripheral nerve injury (*Salvany et al., 2019*; *Herron and Miles, 2012*; *Landoni et al., 2019*), we sought to identify the cellular mechanisms through which this distinct class of cholinergic neurons modulates motor output.

In this study, we employed a combination of genetic, pharmacological and electrophysiological approaches to determine the synaptic mechanisms by which V0c interneurons and their C-boutons modulate motor output. Using Ca$^{2+}$ imaging, we show that populations of lumbar Pitx2[+] interneurons are rhythmically active during fictive locomotion. Chemogenetic inhibition of Pitx2[+] interneurons during fictive locomotion, using designer receptors exclusively activated by designer drugs (DREADDs), decreased the strength of motor output in an M2 receptor-dependent manner. Genetic ablation of V0c interneurons confirmed that C-boutons increase the amplitude, but do not affect rhythmicity, of locomotor output. Chemogenetic excitation of Pitx2[+] interneurons showed that activation of M2 receptors and subsequent regulation of Kv2.1 channels at C-boutons increases motor output by decreasing motoneuron spike half-width and increasing the inter-spike AHP. Together, our findings provide the first direct evidence of the synaptic and cellular mechanisms involved in C-bouton-mediated modulation of spinal locomotor circuitry. Furthermore, we reveal a high degree of specificity within the modulatory control of spinal motor circuits, with discrete modulatory units likely to be devoted to the modulation of specific network output parameters.

## Results

### Pitx2[+] INs are rhythmically active during fictive locomotion

We first confirmed that Pitx2[+] interneurons exhibit rhythmic activity that is phase locked to spinal locomotor network output. Hemisected spinal cords were dissected from *Pitx2::Cre;GCAMP6s* mice to visualize Ca$^{2+}$ activity from groups of Pitx2[+] interneurons whilst simultaneously recording pharmacologically induced locomotor output from L1-L3 lumbar ventral roots (*Figure 1a*). Pitx2[+] interneurons (32 upper lumbar interneurons and 19 lower lumbar interneurons, 6 hemisected spinal cords, 3 mice) exhibited clear rhythmic activity during fictive locomotion (*Figure 1b–c*). The activity of 65.6% of L1-3 interneurons was tightly phase locked with respective L1-L3 ventral root output (indicated by a significant Rayleigh test statistic, p<0.05; *Figure 1d,e*), while the activity of a smaller population of lower lumbar (L4-L6) interneurons was also aligned with upper lumbar output (only 31.6%, *Figure 1d,e*). The remaining cells did not show any phase relationship associated with ventral root

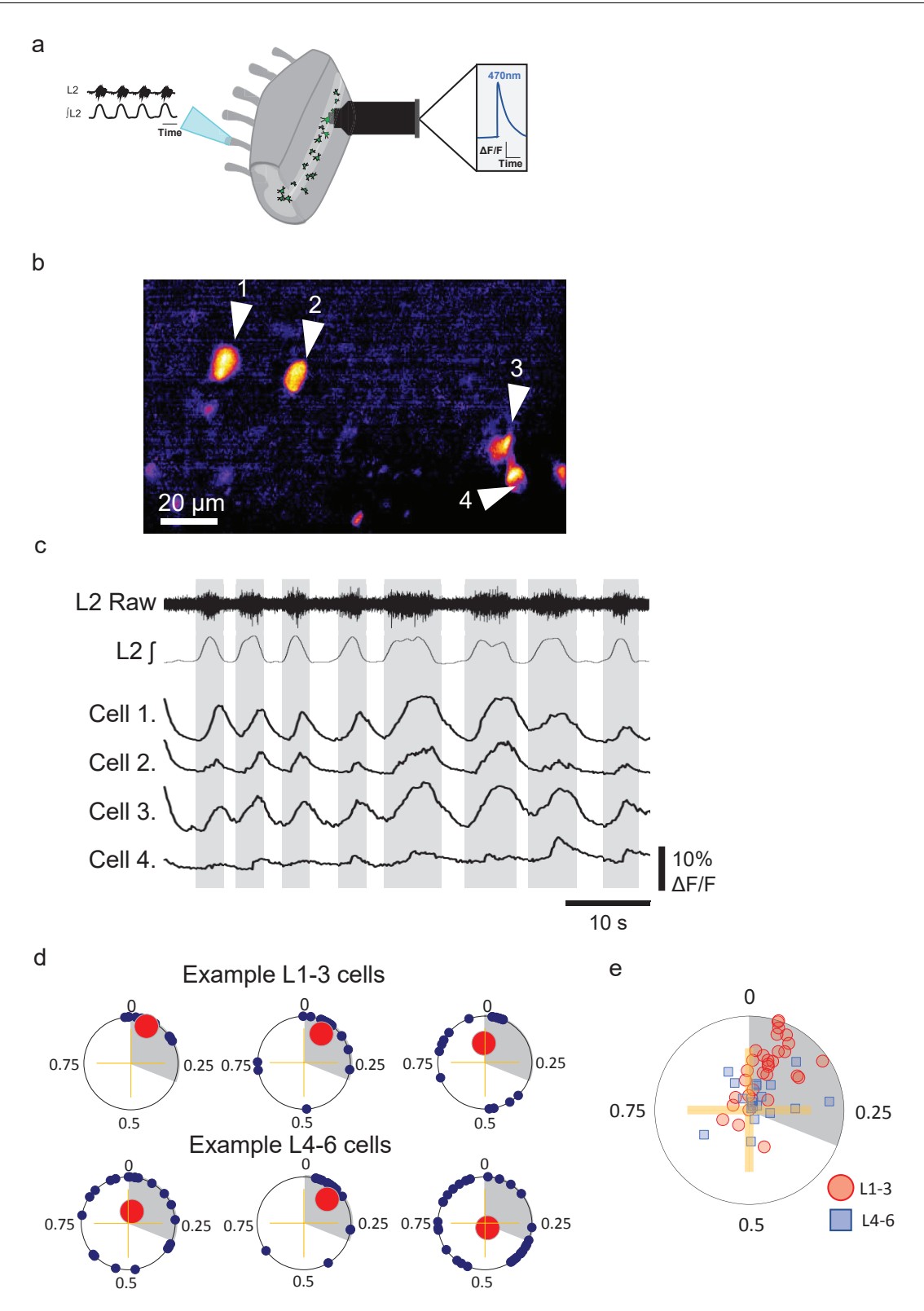

**Figure 1.** Groups of lumbar Pitx2[+] interneurons are rhythmically active during fictive locomotion. (a) Illustration of ventral root recordings and Ca[2+] imaging from Pitx2[+] interneurons (green) in hemisected spinal cords from *Pitx2::Cre;GCAMP6* mice. (b) Pitx2[+] interneurons visualized in the L1-3 upper lumbar regions in a neonatal hemisected spinal cord from a *Pitx2::Cre;GCAMP6* mouse. (c) Ventral root output and examples of Ca[2+] imaging traces from the four neurons marked in (b). (d) Phase plots depicting relationships between Ca[2+] transients in interneurons and ventral root output for three

*Figure 1 continued on next page*

*Figure 1 continued*

neurons in L1-3 (top) and three neurons in L4-6 (bottom). Blue points represent the phasing of individual Ca²⁺ transients within a locomotor cycle, red points represent the mean direction and strength of the phasing for each neuron, and grey shading indicates the average duration of upper lumbar-related bursts. (e) Summary graph of mean direction and strength of phasing for all interneurons analyzed (L1-3, n=31; and L4-6, n=19).

The online version of this article includes the following source data for figure 1:

**Source data 1.** Values for circular phase plot of rhythmically active L1-L3 and L4-L5 Pitx2+ interneurons.

output. These data clearly demonstrate that groups of Pitx2⁺ interneurons are rhythmically active during fictive locomotion and that, as indicated by previous single-cell recordings (*Zagoraiou et al., 2009*), their activity patterns are tightly locked to the locomotor cycle.

## Chemogenetic inhibition of Pitx2⁺ INs decreases the amplitude of locomotor output

We next sought to directly demonstrate the role that Pitx2⁺ interneurons play in controlling moto-neuron output during locomotor network activity. Cre-dependent DREADD mice were used to manipulate the activity of Pitx2⁺ interneurons during fictive locomotion.

We first investigated whether Pitx2⁺ interneurons could be effectively inhibited using DREADD expression. This was assessed by performing whole-cell patch-clamp recordings from spinal cord slices obtained from *Pitx2::Cre;tdTomato;hM4Di* mice. In these mice, Pitx2⁺ cells express both the hM4Di inhibitory receptor and the tdTomato red fluorescent reporter, which allowed them to be identified for single-cell electrophysiology (*Figure 2a*). Pitx2⁺ interneurons, which are distinctively clustered around the central canal, were targeted for recordings (*Zagoraiou et al., 2009*). As previously reported, the majority of Pitx2⁺ interneurons were tonically active at rest (*Zagoraiou et al., 2009*; *Figure 2b,c*). Application of CNO (1 µM) to activate the inhibitory DREADD was found to decrease the tonic firing of Pitx2⁺ interneurons in slices from *Pitx2::Cre;tdTomato;hM4Di* mice (*Figure 2b*, t(8)=6.08, Paired *t*-test, p=0.0003). In slices from *Pitx2::Cre;tdTomato* mice, which express the red fluorescent reporter in Pitx2⁺ interneurons but not the hM4Di receptor, CNO did not change spontaneous firing frequencies (*Figure 2c*, t(4)=0.53, Paired *t*-test, p=0.6228), confirming that the actions of CNO were selective to the hM4Di receptor.

Having established that populations of Pitx2⁺ interneurons are active during fictive locomotion and that their activity can be reduced using the inhibitory DREADD, we went on to assess the effect that chemogenetic inhibition of Pitx2⁺ interneurons has on ventral root output. Ventral root

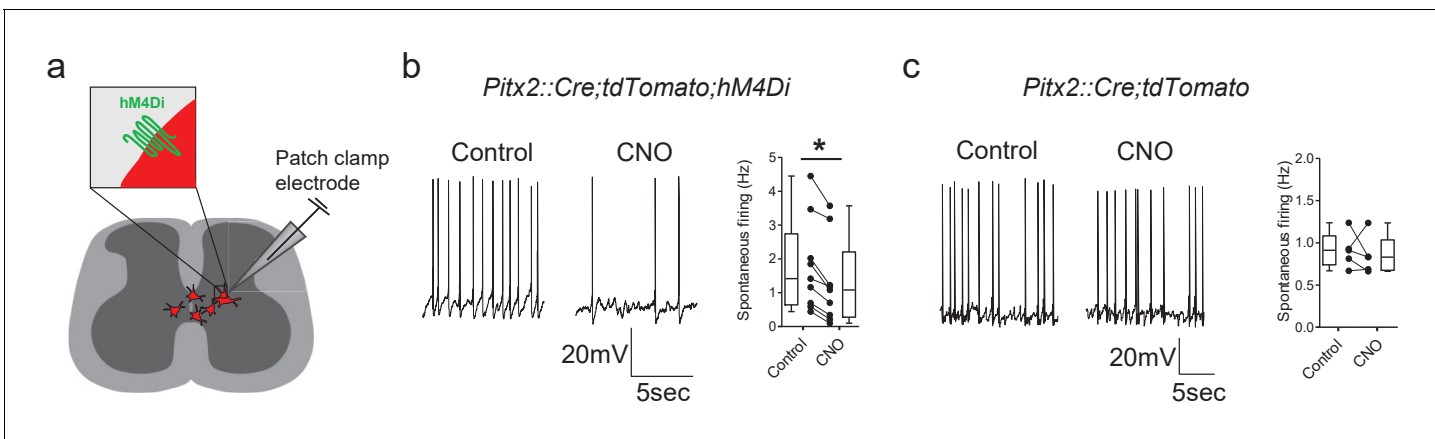

**Figure 2.** The activity of Pitx2+ interneurons can be decreased using DREADDs. (a) Schematic representation of a spinal cord slice containing Pitx2⁺ interneurons (red) targeted for single-cell recordings. (b and c) Spontaneous firing of Pitx2⁺ interneurons from Pitx2::Cre;tdTomato;hM4Di (b) and Pitx2::Cre;tdTomato (c) mice in control conditions and in the presence of CNO (1 µM); along with firing frequency data pooled for all recordings (b, n = 9; c, n = 5). *p<0.05.

The online version of this article includes the following source data for figure 2:

**Source data 1.** Firing frequencies of Pitx2+ interneurons before and after CNO in *Pitx2::Cre;tdTomato;hM4Di* and *Pitx2::Cre;tdTomato* mice.

recordings from L1-L5 were obtained from spinal cords isolated from *Pitx2::Cre;hM4Di* mice. Pitx2[+] interneurons were then inhibited by perfusion of CNO during fictive locomotion (*Figure 3a*). Inhibition of Pitx2[+] interneurons during fictive locomotion did not change burst frequency (Q = 0.57, Friedman's test, p=0.7515; *Figure 3c*, n = 14) or duration (Q = 5.28, Friedman's test, p=0.0712; *Figure 3e*, n = 14) but significantly decreased burst amplitude (−10.7 ± 1.8%, F = 4.34, Repeated measures ANOVA, p=0.0235; *Figure 3g*, n = 14). We also did not observe any significant changes in the variance of burst frequency (F = 4.33, Repeated measures ANOVA, p=0.1146, n = 14) or duration (F = 1.44, Repeated measures ANOVA, p=0.4857, n = 14). CNO had no effect on fictive locomotion in spinal cord preparations obtained from mice that did not express Cre-recombinase (*Figure 3—figure supplement 1*). When M2 receptors were blocked using methoctramine (10 μM; *Figure 3b*), DREADD-mediated inhibition of Pitx2[+] interneurons had no significant effect on burst frequency (F = 1.23, Repeated measures ANOVA, p=0.3203; *Figure 3d*, n = 8), duration (Q = 3.25, Friedman's test, p=0.2359; *Figure 3f*, n = 8) or amplitude (F = 0.89, Repeated measures ANOVA, p=0.4345, *Figure 3h*, n = 8), indicating that the observed reduction in motor output involved C-boutons derived from V0c interneurons.

We next investigated the modulatory effects of Pitx2[+] interneurons on the locomotor-related firing output of individual motoneurons by performing whole-cell patch-clamp recordings from lumbar motoneurons within intact spinal cord, preparations obtained from *Pitx2::Cre;tdTomato;hM4Di* mice (*Figure 4a*). Application of CNO, to chemogenetically inhibit Pitx2[+] interneurons, led to a clear reduction in the rate of motoneuron firing during locomotor bursts (t(3)=5.40, Paired *t*-test, p=0.0125; *Figure 4b*). We also observed an outward current in motoneurons when CNO was applied (102 ± 14 pA, n = 11), which was associated with a decrease in input resistance (control: 88 ± 8 MΩ, CNO: 68 ± 4 MΩ, t(8)=2.455, Paired *t*-test, p=0.0386).

Taken together, these results directly demonstrate that Pitx2[+] interneurons play a specific role in regulating the intensity of locomotor-related motoneuron output and that this modulation is mediated by M2 receptors, known to be expressed at C-bouton synapses. In addition, Pitx2[+] interneurons appear to have little influence on rhythm or pattern generating elements of spinal motor circuits.

## Ablation of V0c INs eliminates muscarinic modulation of the intensity of locomotor output

Results obtained from *Pitx2::Cre;hM4Di* mice indicate that Pitx2[+] interneurons modulate motoneuron output via M2 receptors located at C-bouton synapses. However, chemogenetic inhibition with CNO did not fully abolish Pitx2[+] cell firing (*Figure 2b*). Furthermore, spinal Pitx2[+] interneurons subdivide into cholinergic (V0c) and glutamatergic (V0g) subtypes (*Zagoraiou et al., 2009*), but the transgenic mice utilized in our inhibitory DREADD experiments did not enable the functional dissociation of V0c versus V0g interneurons. Next, we therefore assessed the effects of the specific ablation of V0c interneurons and their C-bouton inputs to motoneurons.

*Pitx2::Cre* mice were crossed with a novel *vAChT-stop-DTA* mouse line (where there are two loxP sequences flanking the stop cassette) in order to achieve specific ablation of V0c interneurons via selective expression of diphtheria toxin A. *Pitx2::Cre;tdTomato;vAChT-stop-DTA* mice were also generated in order to determine the fate of Pitx2[+] neurons at older ages, since our Pitx2 antibody does not work well after P15. We detected V0c interneurons using a combination of antibodies against Pitx2, tdTomato and ChAT, with C-bouton identity further verified by immunoreactivity against vAChT. Immunohistochemical analysis in both neonatal (P2; *Figure 5a* and P7; *Figure 5b–d*) and young adult (P25; *Figure 5e*) transgenic mice demonstrated efficient ablation of Pitx2[+]/ChAT[+] V0c interneurons in *Pitx2::Cre;vAChT-stop-DTA* and *Pitx2::Cre;tdTomato;vAChT-stop-DTA* mice (t(4) =8.42, Unpaired *t*-test, p=0.0011; *Figure 5d*). Furthermore, Pitx2[+]/VAChT[+] C-bouton inputs to motoneurons were also successfully ablated. C-bouton counts from pictures of 50 randomly selected motoneurons from the control group (n = 3 *Pitx2::Cre;tdTomato*) and pictures of 50 randomly selected motoneurons from the experimental group (n = 3 *Pitx2::Cre;tdTomato;vAChT-stop-DTA*) revealed 381 synapses in the control tissue and 16 synapses in the experimental tissue (95.8% deletion). Thus, our genetic strategy led to a highly efficient, conditional ablation.

The effects of genetic ablation of V0c interneurons and their C-boutons were then assessed by investigating muscarinic modulation of pharmacologically induced locomotion recorded from L1-L5 ventral roots from spinal cord preparations from *Pitx2::Cre;vAChT-stop-DTA* mice and wild-type

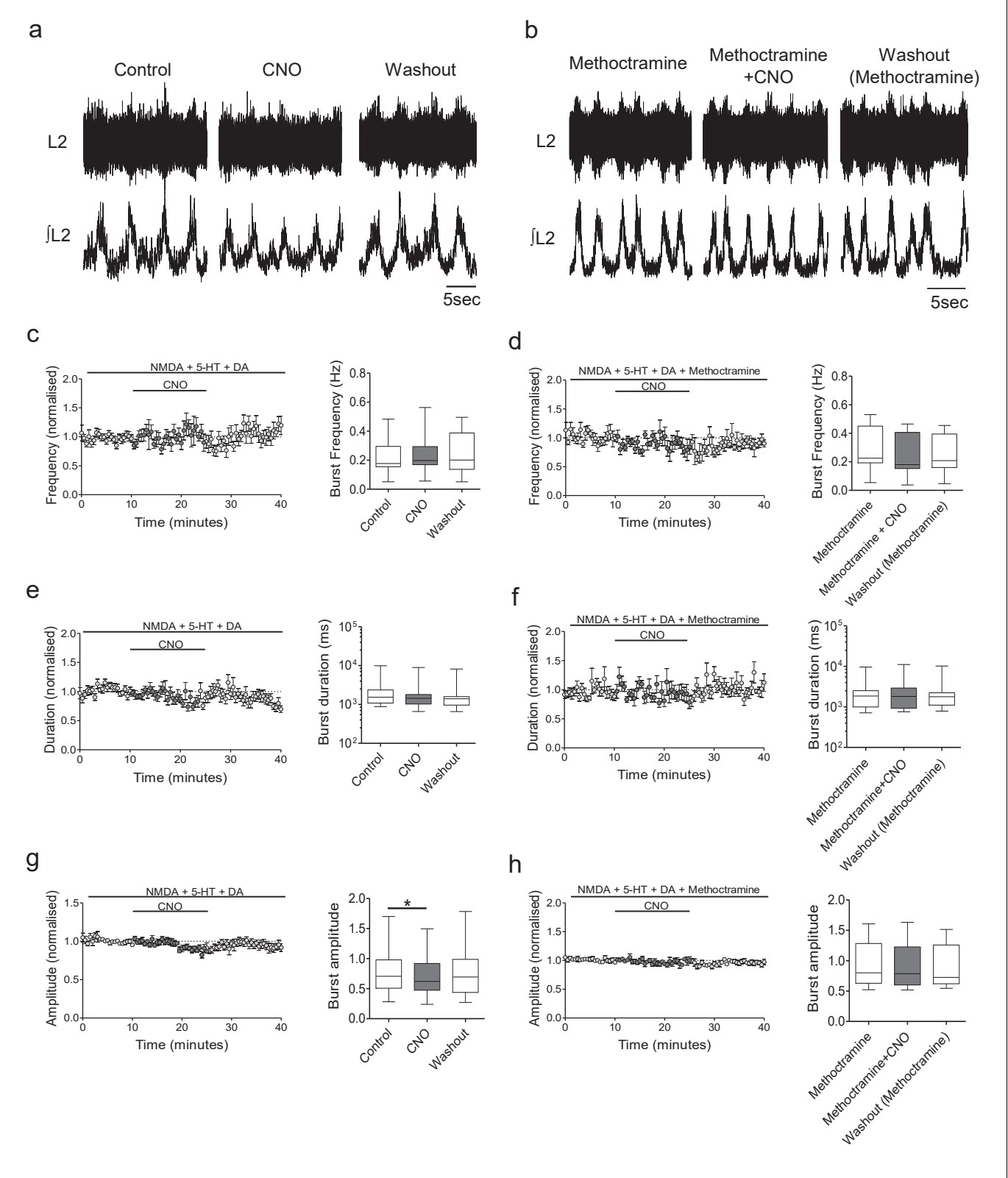

**Figure 3.** Inhibition of Pitx2+ interneurons during fictive locomotion decreases burst amplitude in an M2 receptor-dependent manner. (a and b) Raw (top) and integrated/rectified (bottom) traces illustrating the effects of (a) CNO (1 µM) and (b) CNO plus methoctramine (10 µM) on drug-induced locomotor output recorded from spinal cords of Pitx2::Cre;hM4Di mice. (c – h) Time course plots and box-plots of pooled data showing the effects of CNO alone or CNO plus methoctramine on locomotor burst frequency (b,c), duration (d,e) and amplitude (g,h). *p<0.05.

*Figure 3 continued on next page*

*Figure 3 continued*

The online version of this article includes the following source data and figure supplement(s) for figure 3:

**Source data 1.** Values from ventral root recordings from *Pitx2::Cre;hM4Di* mice in the presence of CNO and CNO co-applied with methoctramine.
**Figure supplement 1.** CNO has no effect on ventral root output in control *hM4Di* mice.
**Figure supplement 1—source data 1.** Values from ventral root recordings from control *hM4Di* mice in the presence of CNO.

animals. In wild-type mice (*Figure 6a*), blockade of M2 receptors via application of methoctramine reduced the amplitude of locomotor-related ventral root bursting (−16.8 ± 4.1%, Q = 9.50, Friedman's test, p=0.0087; *Figure 6g*, n = 12), decreased burst frequency (−11.9 ± 3.6%, Q = 11.62, Friedman's test, p=0.0030; *Figure 6c*, n = 12) and increased burst duration (27.2 ± 3.4%, Q = 20.67, Friedman's test, p<0.0001; *Figure 6e*, n = 12), similar to recently reported effects of M2 receptor antagonists on whole cord output in neonatal mice (*Nascimento et al., 2019*). Blockade of M2 receptors in *Pitx2::Cre;vAChT-stop-DTA* mice (*Figure 6b*) also led to a decrease in burst frequency (−24.4 ± 7.6%, F = 16.59, Repeated measures ANOVA, p=0.0099; *Figure 6d*, n = 10) and an increase in burst duration (68.34 ± 23.7%, Q = 9.80, Friedman's test, p=0.0063; *Figure 6f*, n = 10). However, methoctramine no longer had an effect on burst amplitude when V0c interneurons and C-boutons were ablated (2.3 ± 2.2%, F = 0.32, Friedman's test, p=0.7263; *Figure 6h*, n = 10). These data demonstrate that V0c interneurons and their C-bouton contacts with motoneurons are solely responsible for the M2 muscarinic receptor-mediated modulation of the intensity of locomotor-related output. In addition, these results demonstrate that additional effects of M2 receptor activation on locomotor network function do not involve V0c interneurons or C-bouton synapses.

## Chemogenetic excitation of Pitx2[+] INs reveals synaptic mechanisms at C-boutons

To reveal the cellular mechanisms underlying C-bouton-mediated modulation of motor output, we used excitatory (hM3Dq receptor) DREADD expression in Pitx2[+] interneurons to activate C-boutons while recording from lumbar motoneurons in intact spinal cord preparations. We first tested whether Pitx2[+] interneurons could be excited using this DREADD approach. Application of CNO reliably

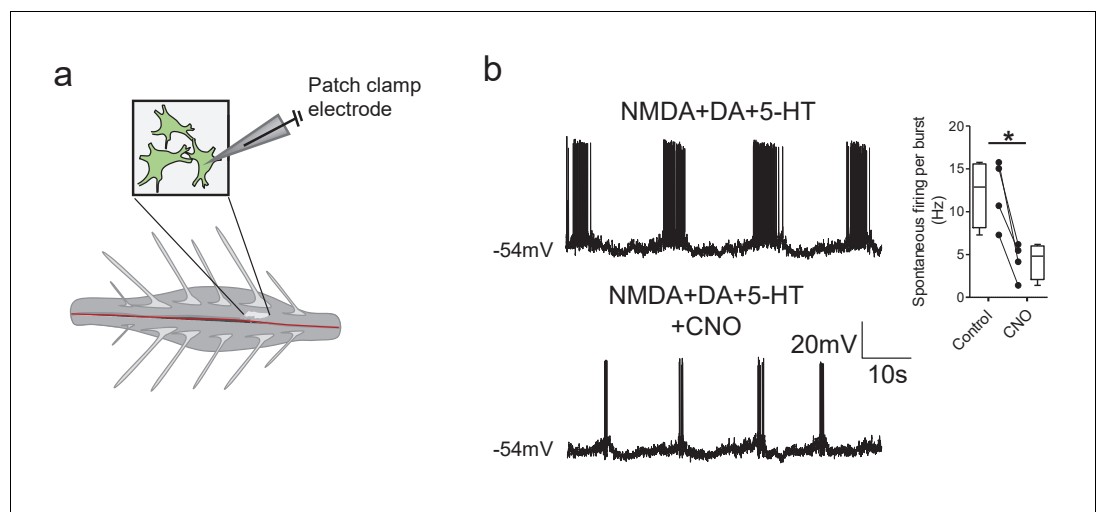

**Figure 4.** DREADD-mediated inhibition of Pitx2[+] interneurons decreases motoneuron firing rates during fictive locomotion. (a) Schematic depicting the intact spinal cord preparation used for patch clamp recordings of lumbar motoneurons (green) during fictive locomotion. (b) Spontaneous firing recorded from a motoneuron during pharmacologically induced fictive locomotion in control conditions and after the addition of CNO; along with firing frequency data pooled for all recordings (n = 4); *p<0.05.
The online version of this article includes the following source data for figure 4:

**Source data 1.** Motoneuron firing frequency, input resistance and changes in holding current by CNO during fictive locomotion in *Pitx2::Cre;hM4Di* mice.

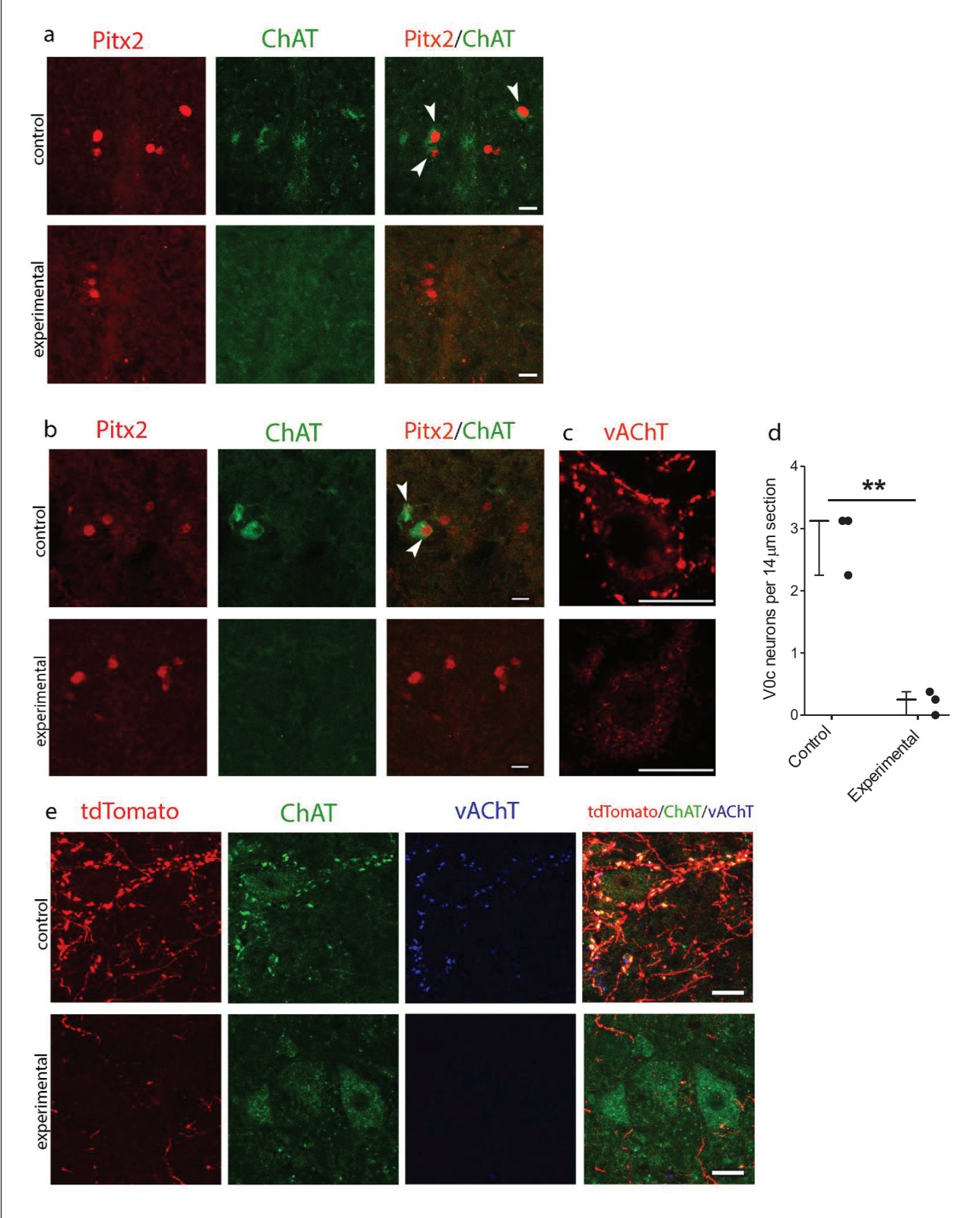

**Figure 5.** Genetic ablation of cholinergic Pitx2⁺ interneurons (V0c) eliminates C-boutons around motoneuron somata. (a) Immunofluorescence in spinal cord sections of P2 *wt* (control) and *Pitx2::Cre;vAChT-stop-DTA* (experimental) mice using antibodies against Pitx2 (red) and Choline Acetyl Transferase (ChAT, green). (b) Immunofluorescence in spinal cord sections of P7 upper lumbar levels from *wt* (control) and *Pitx2::Cre;;vAChT-stop-DTA* (experimental) mice using antibodies against Pitx2 (red) and ChAT (green). White arrows in (a) and (b) point to double positive neurons. Neurons that

*Figure 5 continued on next page*

Figure 5 continued

are positive for cholinergic markers only but not positive for Pitx2 are preganglionics, typically present in the intermediate zone in these levels (**a**, **b**). (**c**) Immunofluorescence of motoneurons and their C-bouton terminals in upper lumbar spinal cord sections using an antibody against vAChT. (**d**) Average number of V0c neurons per 14 μm section in control and experimental P7 mice. (**e**) Immunofluorescence of motoneurons and their C-bouton terminals in upper lumbar spinal cord sections of P25 *Pitx2::Cre;tdTomato* (control) and *Pitx2::Cre;tdTomato;vAChT-stop-DTA* (experimental) mice using antibodies against tdTomato (red), ChAT (green) and vAChT (blue); Photos acquired with confocal microscopy using (**a**, **b**) 20x lens and (**c**, **e**) 40x lens. Sections thickness was 14 μm and scale bar is 20 μm; n = 3 mice for each condition; **p<0.01.

The online version of this article includes the following source data for figure 5:

**Source data 1.** Counts of V0c interneurons per 14 μm section in P7 wt and *Pitx2::Cre;vAChT-stop-DTA* mice.

increased the spontaneous firing frequency of Pitx2$^+$ interneurons clustered near the central canal in spinal cord slices prepared from Pitx2::Cre;tdTomato;hM3Dq mice (t(7)=2.45, Paired *t*-test, p=0.0441; *Figure 7a*).

Having established that Pitx2$^+$ interneurons from *Pitx2::Cre;hM3Dq* mice can be excited via CNO application, we next performed whole-cell patch-clamp recordings from motoneurons to directly examine the cellular effects of activating Pitx2$^+$ interneurons. We began by assessing whether activation of Pitx2$^+$ interneurons induced any subthreshold currents that might adjust the resting membrane potential of motoneurons (*Figure 7b*). In spinal cords from *Pitx2::Cre;hM3Dq* mice, activation of Pitx2$^+$ interneurons with CNO elicited an inward current in motoneurons (−37 ± 6 pA, n = 14; *Figure 7b* left) that was associated with an increase in input resistance (control: 77 ± 9 MΩ, CNO: 90 ± 12 MΩ, W=-46, Wilcoxon signed-rank test, p=0.0217,n = 10). The M2 receptor antagonist methoctramine, blocked both this inward current (−2 ± 4 pA, n = 9; *Figure 6b*, middle) and the change in input resistance (methoctramine: 82 ± 13 MΩ, methoctramine+CNO: 89 ± 16 MΩ, t(7) =1.68, Paired *t*-test, p=0.1368, n = 8), supporting that these CNO-induced effects involve V0c interneurons and their C-boutons. Given that Kv2.1 channels are expressed by motoneurons and exhibit clustering at C-bouton synapses (*Wilson et al., 2004*; *Deardorff et al., 2014*), they could be involved in V0c interneuron-mediated modulation of motor output. To address whether changes in the subthreshold properties of motoneurons following activation of Pitx2$^+$ interneurons involves Kv2.1 channels, we perfused CNO in the presence of the selective Kv2.1 channel blocker guangxitoxin-1E (50 nM). At this concentration, guangxitoxin-1E is thought to be highly selective for Kv2.1 channels, with limited impact on other channels including Kv4 channels (*Herrington et al., 2006*; *Liu and Bean, 2014*; *Fletcher et al., 2017*). Co-application of CNO with guangxitoxin-1E still induced an inward current (−30 ± 7 pA, n = 10; *Figure 7b*, right) that was associated with an increase in input resistance (guangxitoxin-1E: 49 ± 5 MΩ, guangxitoxin-1E and CNO: 58 ± 3 MΩ, t(7) =3.28, Paired *t*-test, p=0.0135, n = 8), indicating that Kv2.1 channels are not involved in this response.

We next assessed whether activation of Pitx2$^+$ interneurons modulates the firing output of motoneurons in *Pitx2::Cre;hM3Dq* mice. Motoneuron input-output relationships were investigated using a series of depolarizing current steps applied in current-clamp mode during whole-cell patch-clamp recordings of motoneurons in isolated spinal cord preparations. Activation of Pitx2$^+$ interneurons with CNO (*Figure 7c*) decreased motoneuron rheobase (t(22)=3.32, Paired *t*-test, p=0.0031), increased the current required to reach a depolarizing block of firing (W=-148, Wilcoxon signed-rank test, p=0.0054) and increased the maximum firing rate of motoneurons (t(23)=4.13, Paired *t*-test, p=0.0004). When M2 receptors were blocked with methoctramine (*Figure 7d*) activation of Pitx2$^+$ interneurons had no effect on motoneuron rheobase (t(10)=0.04, Paired *t*-test, p=0.9709), depolarizing block (t(10)=0.000, Paired *t*-test, p=1.0000) or maximum firing rates (W = 19.00, Wilcoxon signed-rank test, p=0.3552). Activation of Pitx2$^+$ interneurons in the presence of the Kv2.1 blocker guangxitoxin-E, still reduced motoneuron rheobase (t(13)=2.47, Paired *t*-test, p=0.0279), but no longer affected the current required for a depolarizing block (W = −21.00, Wilcoxon signed-rank test, p=0.3532) nor motoneuron maximum firing rates (t(13)=0.000, Paired *t*-test, p=1.0000). These results demonstrate that chemogenetic activation of Pitx2$^+$ interneurons increases the firing output of motoneurons through activation of M2 receptors and subsequent regulation of Kv2.1 channels that are present at C-bouton synapses.

The observed decrease in rheobase could indicate a change in the firing threshold of motoneurons. To address this, a depolarizing current ramp (1 s duration) was used to measure action potential

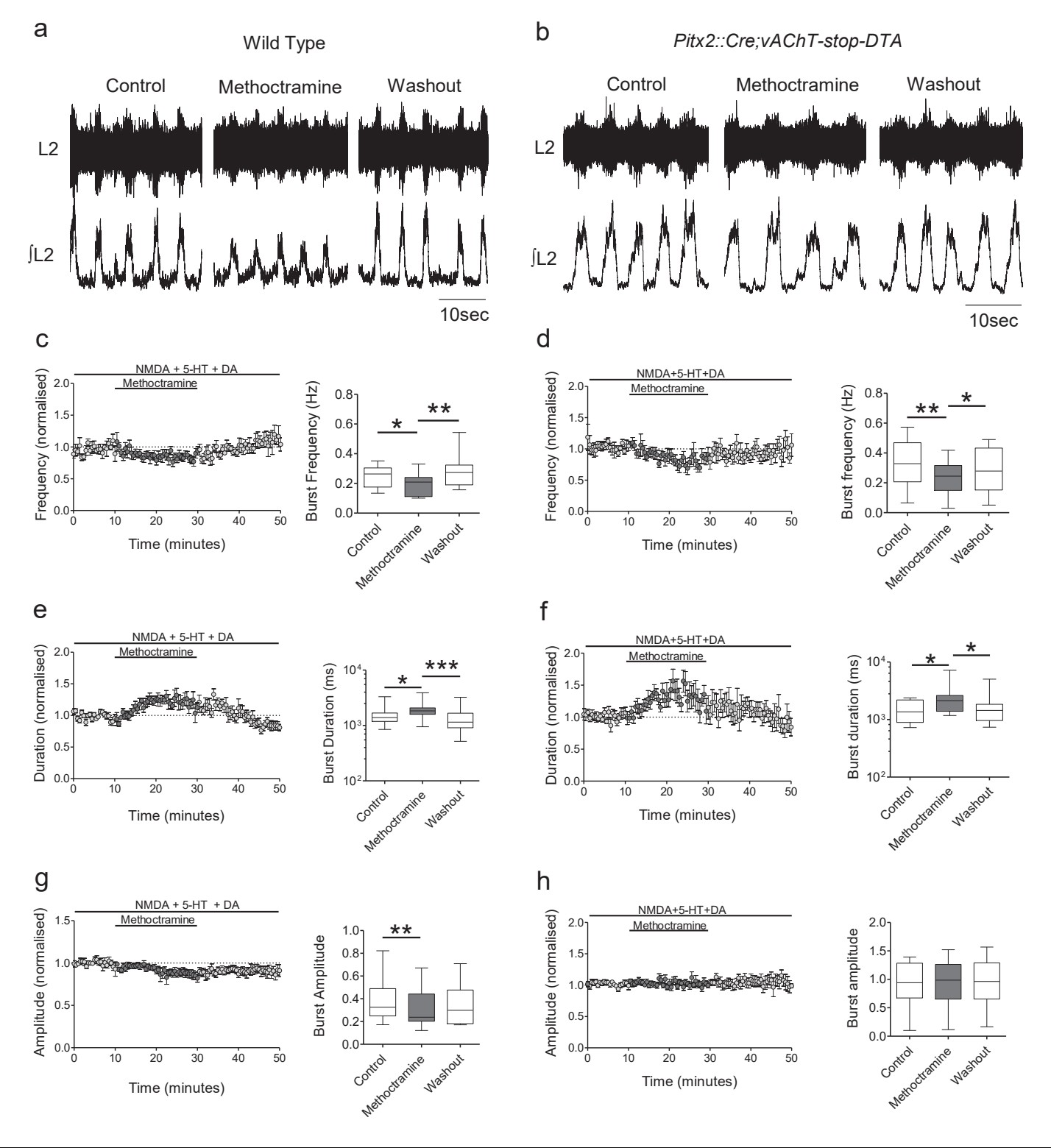

**Figure 6.** Genetic ablation of cholinergic Pitx2+ interneurons (V0c) removes M2 receptor-dependent modulation of locomotor burst amplitude. (**a, b**) Raw (top) and integrated/rectified (bottom) traces with averaged time course plots and mean pooled data illustrating the effects of methoctramine (10 µM) on Wild Type (n = 12) and *Pitx2::Cre;vAChT-stop-DTA* (n = 10) mice lumbar ventral root burst frequency (**c, d**), duration (**e, f**) and amplitude (**g, h**); *p<0.05, **p<0.01, ***p<0.001.

The online version of this article includes the following source data for figure 6:

**Source data 1.** Values from ventral root recordings from WT and *Pitx2::Cre;;vAChT-stop-DTA* mice.

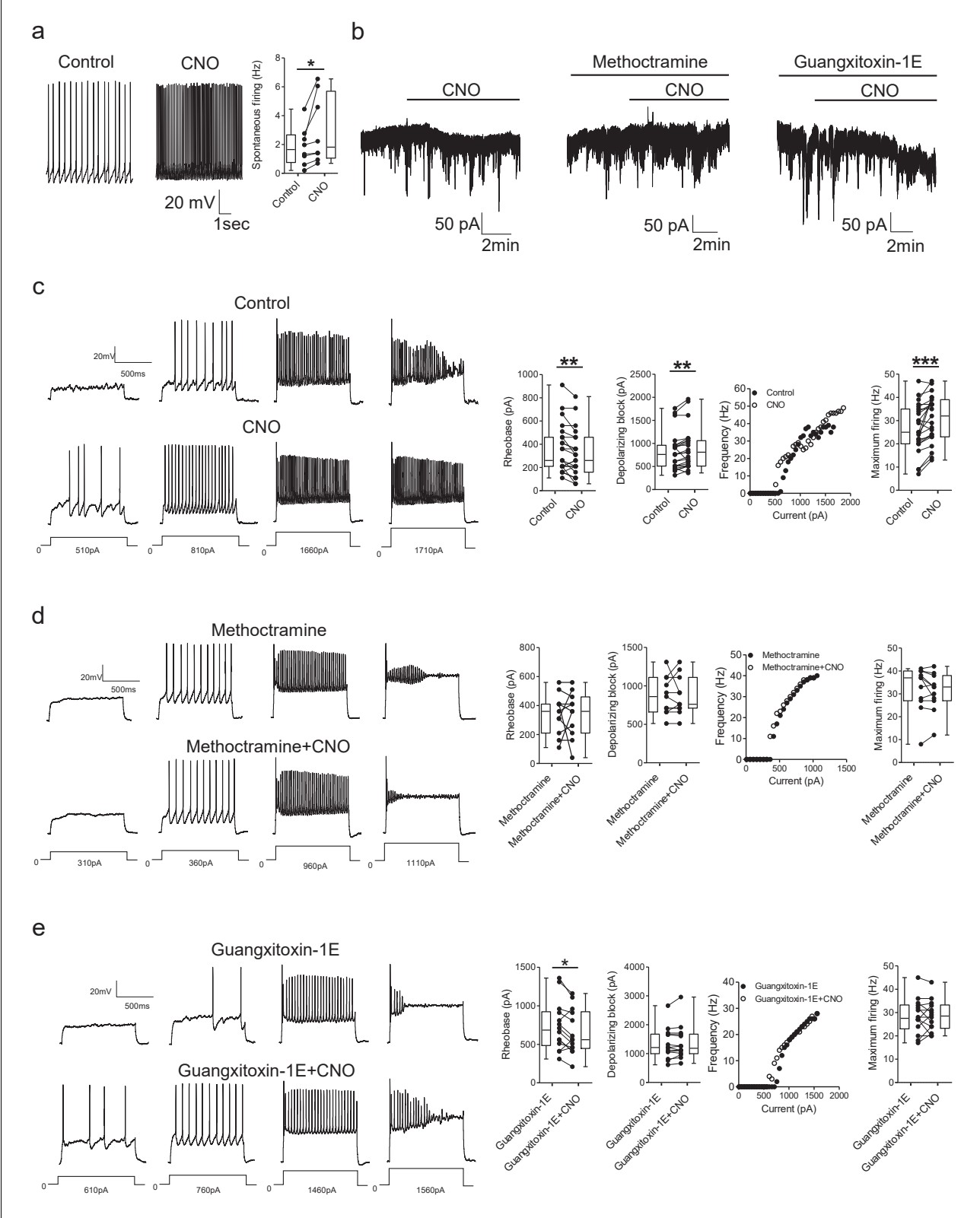

**Figure 7.** DREADD-based activation of Pitx2[+] interneurons increases motoneuron firing via M2 receptors and Kv2.1 channels. (**a**) Spontaneous firing of a Pitx2[+] interneuron from a *Pitx2::Cre;tdTomato;hM3Dq* mouse in control conditions and during the application of CNO; along with firing frequency data pooled for all recordings (n = 8). (**b**) Representative traces illustrating changes in holding current (Vhold = −60 mV) during recordings of motoneurons in the presence of CNO alone (left) or co-applied with methoctramine (10 μM, middle) or guangxitoxin-1E (50 nM, right). (**c**) Motoneuron

*Figure 7 continued*

firing in response to current steps in control conditions and in the presence of CNO (left) with pooled data plotted to show changes in rheobase, depolarizing block, current-frequency relationships and maximum firing (right) (n = 23). (d, e) Examples of motoneuron firing and pooled data depicting firing parameters in the presence of methoctramine and methoctramine co-applied with CNO (*d*; n = 11) or Guangxitoxin-1E and CNO co-applied with Guangxitoxin-1E (*e*; n = 14). *p>0.05, **p<0.01, ***p<0.001.

The online version of this article includes the following source data and figure supplement(s) for figure 7:

**Source data 1.** Values for firing of Pitx2+ interneurons with CNO in *Pitx2::Cre;tdTomato;hM3Dq* mice and values for holding current, rheobase, depolarizing block and maximum firing in motoneurons from *Pitx2::Cre;hM3Dq* mice.
**Figure supplement 1.** CNO has no effect on motoneuron properties in *hM3Dq* mice.
**Figure supplement 1—source data 1.** Values for changes in holding current, rheobase, depolarizing block, maximum firing and mAHP amplitude in motoneurons from control *hM3Dq* mice.

threshold. Activation of Pitx2$^+$ interneurons with CNO decreased motoneuron firing threshold measured from ramps (W = 62.00, Wilcoxon signed-rank test, p=0.0029; *Figure 8a*), an effect which was blocked by methoctramine (t(3)=0.67, Paired *t*-test, p=0.5513; *Figure 8b*), but not by guangxitoxin1-E (t(5)=3.10, Paired *t*-test, p=0.0362; *Figure 8c*). These data indicate that activation of M2 receptors at C-bouton synapses reduces the current required for motoneuron firing by hyperpolarizing the action potential threshold via Kv2.1-independent mechanisms.

Previous, indirect evidence has suggested that activation of M2 receptors at C-bouton synapses is likely to increase motoneuron output through a reduction in the amplitude of the mAHP (*Miles et al., 2007*). We therefore tested this directly using our chemogenetic approach. Single action potentials were evoked from motoneurons in current-clamp mode using brief (10 ms) depolarizing current steps to assess whether activation of Pitx2$^+$ interneurons reduced the action potential mAHP within isolated spinal cord preparations from *Pitx2::Cre;hM3Dq* mice. Surprisingly, activation of Pitx2$^+$ interneurons with CNO increased the mAHP amplitude (W = 204, Wilcoxon signed-rank test, p=0.0010; *Figure 8d*). This effect was blocked when CNO was co-applied with either the M2 receptor antagonist methoctramine (t(6)=0.63, Paired *t*-test, p=0.5513; *Figure 8e*) or the Kv2.1 channel blocker guangxitoxin1-E (t(15)=1.27, Paired *t*-test, p=0.2208; *Figure 8f*).

Kv2.1 channels can contribute to the inter-spike afterhyperpolarization, which can facilitate greater firing rates by enabling sodium channels to recover from inactivation between spikes (*Liu and Bean, 2014*; *Johnston et al., 2008*). Thus, the changes we observed in the mAHP, the amount of current required to induce a depolarizing block and the maximum firing rates of motoneurons when Pitx2$^+$ interneurons are activated could reflect modulation of Kv2.1 conductances at C-bouton synapses. To evaluate this, we calculated the magnitude of the inter-spike AHPs that followed the last five action potentials induced by current steps that elicited maximum firing frequencies in motoneurons from *Pitx2::Cre::hM3Dq* mice. We observed an increase in the amplitude of inter-spike AHPs in motoneurons following activation of Pitx2$^+$ interneurons with CNO (t(20)=8.44, Paired *t*-test, p<0.0001; *Figure 8g*). This effect was blocked by both the M2 receptor antagonist methoctramine (t(9)=1.02, Paired *t*-test, p=0.3354; *Figure 8h*) and the Kv2.1 blocker guangxitoxin1-E (t(11)=1.45, Paired *t*-test, p=0.1736; *Figure 8i*). These data indicate that C-bouton activation enables motoneurons to fire during sustained and intense stimulation by increasing the inter-spike AHP.

Kv2.1 channels have also been shown to affect action potential duration which can contribute to changes in motoneuron firing by allowing faster, sharper spikes (*Fletcher et al., 2017*). To address if C-bouton activation also modulates motoneuron firing rates by controlling spike duration, we examined spike half-width measured from single action potentials evoked in motoneurons from *Pitx2::Cre::hM3Dq* mice. Activation of Pitx2$^+$ interneurons with CNO decreased spike half-width (W = 182, Wilcoxon signed-rank test, p=0.0007; *Figure 8j*). This effect was blocked by both the M2 receptor antagonist methoctramine (W=-42, Wilcoxon signed-rank test, p=0.0674; *Figure 8k*) and the Kv2.1 blocker guangxitoxin1-E (t(12)=1.10, Paired *t*-test, p=0.2951; *Figure 8l*). These observations demonstrate that M2 receptor-mediated modulation of Kv2.1 channels at C-bouton synapses also facilitates greater motoneuron firing rates by reducing spike half-width.

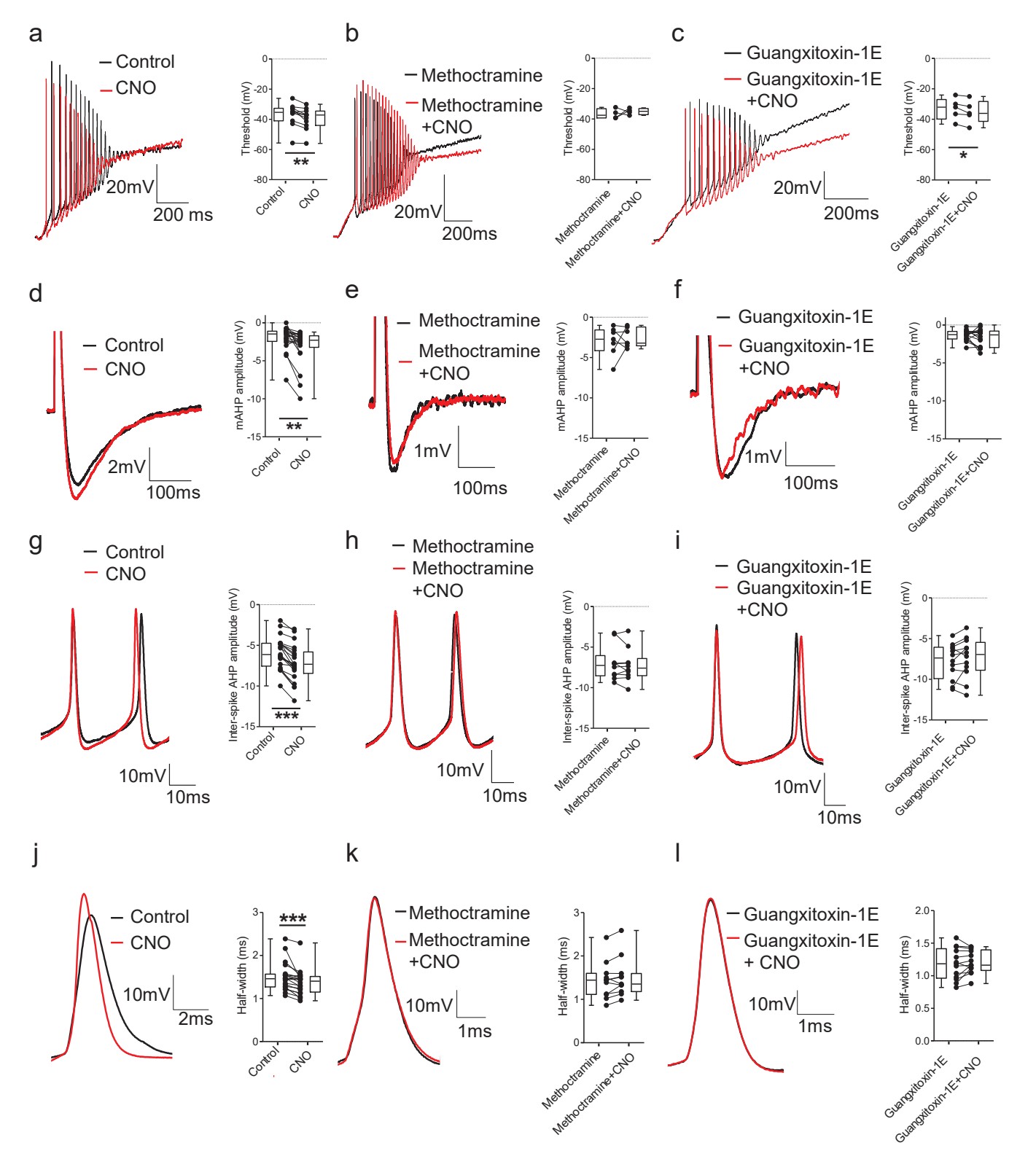

**Figure 8.** DREADD-based activation of Pitx2[+] interneurons influences action potential parameters in motoneurons. (**a–c**) Motoneuron firing in response to depolarizing current ramps (1 s duration) in whole spinal cord preparations from *Pitx2::Cre;hM3Dq* mice demonstrating action potential threshold following activation of Pitx2[+] interneurons with CNO alone (*a*; 1 µM, a; n = 12), and CNO co-applied with either methoctramine (10 µM) (*b*; n = 4) or guangxitoxin-1E (50 nM) (*c*; n = 5). (**d–f**) Truncated single action potentials illustrating the amplitude of the mAHP following application of CNO alone

*Figure 8 continued on next page*

Figure 8 continued

(*d*; n = 22), and CNO co-applied with either methoctramine (*e*; n = 7) or guangxitoxin-1E (*f*; n = 11). (**g–i**) Successive action potentials recorded during repetitive firing showing inter-spike AHP amplitude following application of CNO alone (*g*; n = 21), and CNO co-applied with either methoctramine (*h*; n = 10) or guangxitoxin-1E (*i*; n = 12). (**j–l**) Recordings of single action potentials illustrating action potential half-width following application of CNO alone (*j*; n = 20), and CNO in the presence of either methoctramine (*k*; n = 11) or guangxitoxin-1E (*l*; n = 13). *p<0.05, **p<0.01, ***p<0.001. The online version of this article includes the following source data for figure 8:

**Source data 1.** Values for threshold, mAHP, inter-spike AHP and half-width in motoneurons from *Pitx2::Cre; hM3Dq* mice.

## Discussion

In this study, we interrogated the roles and cellular mechanisms of cholinergic modulation within motor circuits by focusing on a uniquely discrete population of genetically identifiable cholinergic spinal interneurons marked by expression of the transcription factor Pitx2. By combining $Ca^{2+}$ imaging, chemogenetic manipulation and conditional ablation of cholinergic $Pitx2^+$ interneurons (V0c), we demonstrate that recruitment of V0c interneurons and their C-bouton synapses leads to activation of postsynaptic M2 receptors, which in turn regulate Kv2.1 channel function to facilitate increased and sustained motoneuron firing. Thus, activation of the V0c system results in augmented motor output that will translate into stronger muscle activity. Such intrinsic modulation will enable spinal motor output, and muscular contraction, to be matched to variable behavioral demands.

Previous single cell recordings from $Pitx2^+$ interneurons during pharmacologically induced locomotor-related activity demonstrated that the majority of $Pitx2^+$ cells exhibit bursts of activity in phase with segmentally aligned motoneuron targets (*Zagoraiou et al., 2009*). We have now extended these analyses to a population level by imaging $Ca^{2+}$ fluctuations in groups of lumbar $Pitx2^+$ interneurons during locomotor network activity. We find that a high proportion of $Pitx2^+$ interneurons exhibit rhythmic activity that is phase locked to bursts of locomotor-related activity recorded from ventral roots. Similar to previous work (*Zagoraiou et al., 2009*), we found that the strength of the relationship between $Pitx2^+$ cell activity and locomotor output was more robust in upper compared with lower lumbar regions. This may reflect rostro-caudal differences in the density of V0c and V0g interneurons in the lumbar spinal cord (*Zagoraiou et al., 2009*; *Enjin et al., 2010*).

Following the demonstration that $Pitx2^+$ interneurons are rhythmically active during locomotor network activity, we next investigated their modulatory actions on locomotor-related output by using Cre-dependent expression of DREADDs to manipulate their activity. We first verified the effectiveness and specificity of our DREADD-based approach by showing that single-cell properties and baseline network output did not differ between DREADD and control mice prior to the application of CNO (*Tables 1*, *2* and *3*, *Supplementary file 1*). Both high doses of CNO (10 µM) and the CNO derivative clozapine, which is only likely to be produced at significant levels when administered in vivo (*MacLaren et al., 2016*; *Pirmohamed et al., 1995*), can exhibit off-target effects on endogenous receptors, including 5-HT receptors known to contribute to the control of locomotor networks (*Armbruster et al., 2007*; *Gomez et al., 2017*). We therefore also verified that CNO (1 µM) had no detectable effects on ventral root output, single motoneuron firing, or subthreshold motoneuron properties when applied to preparations obtained from animals that do not express DREADD receptors (*Figure 3—figure supplement 1* and *Figure 7—figure supplement 1*). Having verified our

**Table 1.** Intrinsic properties of lumbar motoneurons from *hM3Dq* and *Pitx2::Cre; hM3Dq* mice.

|  | *hM3Dq* | *Pitx2::Cre; hM3Dq* |
|---|---|---|
| Resistance (MΩ) | 63 ± 3 | 58 ± 4 |
| Capacitance (pF) | 114 ± 7 | 113 ± 4 |
| Membrane potential (mV) | −60 ± 2 | −62 ± 2 |
| Rheobase (pA) | 263 ± 43 | 353 ± 43 |
| Depolarizing block (pA) | 666 ± 44 | 817 ± 84 |
| Maximum firing (Hz) | 26 ± 2 | 27 ± 2 |
|  | n = 16 | n = 23 |

**Table 2.** Intrinsic properties of Pitx2[+] interneurons from control, excitatory and inhibitory DREADD mice.

| | Pitx2::Cre;tdTomato | Pitx2::Cre;tdTomato;hM4Di | Pitx2::Cre;tdTomato;hM3Dq |
|---|---|---|---|
| Resistance (MΩ) | 336 ± 48 | 307 ± 30 | 320 ± 16 |
| Capacitance (pF) | 28 ± 2 | 35 ± 3 | 26 ± 4 |
| Membrane potential (mV) | −52 ± 3 | −52 ± 2 | −53 ± 1 |
| Spontaneous firing (Hz) | 2.2 ± 0.6 | 1.8 ± 0.5 | 2.0 ± 0.4 |
| | n = 11 | n = 9 | n = 9 |

DREADD approach, we first assessed the effects of inhibiting Pitx2[+] interneurons on ongoing locomotor-related activity. DREADD-mediated inhibition revealed that Pitx2[+] interneurons modulate the amplitude of locomotor-related output in an M2 muscarinic receptor-dependent manner. This finding is consistent with previous reports of a reduction in the amplitude of locomotor-related bursts when applying antagonists of M2-type muscarinic receptors (*Miles et al., 2007*; *Nascimento et al., 2019*). We next addressed whether cholinergic V0c interneurons, and their C-bouton inputs to motoneurons, were solely responsible for the modulation of motor output we observed. In preparations from animals in which V0c cells had been genetically ablated, blockade of M2 receptors no longer affected the amplitude of locomotor-related output. Thus, Pitx2[+] interneuron-dependent modulation of the intensity of motor output originates from cholinergic V0c interneurons and their C-bouton synapses on motoneurons. Despite previous evidence of a low density of V0c-derived synapses in the intermediate zone of the spinal cord (*Zagoraiou et al., 2009*), our findings demonstrate a primary role for V0c interneurons in the control of motoneuron output with little or no impact on locomotor CPG interneurons. Furthermore, although glutamatergic V0g interneurons are also known to project to the intermediate zone as well as the dorsal horn of the spinal cord (*Zagoraiou et al., 2009*), the fact that locomotor-related activity was unaltered when Pitx2[+] cell activity was manipulated in the presence of muscarinic receptor blockers indicates that V0g interneurons do not contribute to the generation or modulation of locomotor network output and hence their functional roles remain elusive.

Recent studies have reported a diversity of muscarinic receptor-mediated effects on spinal locomotor circuits, as revealed by modulation of the frequency, duration and amplitude of locomotor-related output (*Nascimento et al., 2019*; *Jordan et al., 2014*; *Finkel et al., 2014*). In keeping with these observations, when V0c interneurons were genetically ablated, pharmacological blockade of M2 receptors revealed V0c-independent muscarinic modulation of the frequency and duration of locomotor bursts. Given that M2 receptors are not only juxtaposed to C-boutons but are also widely distributed throughout motoneuron somata (*Wilson et al., 2004*; *Miles et al., 2007*; *Hellström et al., 2003*; *Welton et al., 1999*) and on spinal interneurons (*Höglund and Baghdoyan, 1997*), it is likely that additional, as yet genetically undefined, neuronal sources of intraspinal acetylcholine (*Bertrand and Cazalets, 2011*; *Gillberg et al., 1988*; *Gillberg et al., 1990*; *Sherriff and Henderson, 1994*) are responsible for these M2 receptor-dependent changes in the rhythm and pattern of locomotor output. Given the specific nature of the effects of V0c interneurons on spinal motor circuits, it will be interesting to investigate whether similarly discrete subsets of spinal cells fulfil the other modulatory roles uncovered by blockade of endogenous cholinergic signalling. Recent work suggests that such separate modulatory pathways may act in concert, via different muscarinic receptor subtypes, to ensure balanced motor circuit output (*Nascimento et al., 2019*).

**Table 3.** Ventral root burst frequency and duration measured in the presence of NMDA, 5-HT and DA from preparations obtained from *hM4Di* and *Pitx2::Cre;hM4Di* mice.

| | hM4Di | Pitx2::Cre;hM4Di |
|---|---|---|
| Burst frequency (Hz) | 0.23 ± 0.03 | 0.21 ± 0.03 |
| Burst duration (ms) | 2049 ± 507 | 2393 ± 554 |
| | n = 14 | n = 19 |

Inhibition of Pitx2$^+$ interneurons during locomotor network activity was associated with both a reduction in population level motoneuron output, measured via ventral root recordings, and a reduction in motoneuron firing rates, evidenced by patch-clamp recordings from individual cells. We next took advantage of our genetic access to this specific population of interneurons, and our successful DREADD-based manipulation of their activity, to directly identify the synaptic and cellular mechanisms underlying the discrete modulatory pathway mediated by V0c interneurons.

DREADD-mediated excitation of V0c interneurons was associated with an M2 receptor-dependent increase in the excitability of motoneurons. This increased excitability was characterized by a decrease in rheobase due to hyperpolarization of the action potential threshold, greater resistance to depolarizing block, which allowed motoneurons to fire in response to greater amplitude stimuli, and an increase in maximum firing rates. Potential mediators of these effects include postsynaptic Kv2.1 potassium channels and SK-type calcium-dependent potassium channels, which are both clustered at C-boutons synapses (*Wilson et al., 2004*; *Hellström et al., 2003*; *Muennich and Fyffe, 2004*). Blockade of Kv2.1 channels during activation of V0c interneurons prevented the increased resistance to depolarizing block and the rise in maximum firing frequency, but not hyperpolarization of the action potential threshold. Given that recent modeling experiments indicate that blockade of SK channels depolarizes the action potential threshold in midbrain dopamine neurons (*Iyer et al., 2017*), SK channels at C-bouton synapses may also play a role in controlling the spike threshold of motoneurons. This is supported further by previous work indicating that motoneurons that express the SK3 channel isoform might have reduced rheobase when compared to motoneurons that do not express this isoform (*Deardorff, 2013*). In contrast, the other mechanisms leading to increased motoneuron excitability following activation of V0c interneurons appear to involve modulation of Kv2.1 channels, which are known to facilitate repetitive firing in a wide range of neuronal types (*Liu and Bean, 2014*; *Malin and Nerbonne, 2002*; *Guan et al., 2013*) and to play an important role in the repolarization phase of action potentials in motoneurons (*Romer et al., 2019*; *Gao and Ziskind-Conhaim, 1998*).

Our data, which support a predominant role for Kv2.1 channels in C-bouton-mediated modulation of motoneuron excitability, contrast previous work, which suggested modulation of SK channels and a reduction in the mAHP were responsible for increased motoneuron output upon C-bouton activation (*Miles et al., 2007*). However, this previous work relied on broad pharmacological manipulation of muscarinic receptors in spinal cord slices, prior to the discover of the C-bouton source cells, and therefore likely activated a wide population of receptors beyond those localized at C-bouton synapses. Given that activation of muscarinic receptors has a number of different effects on motoneurons (*Nascimento et al., 2019*), it is plausible that M2 receptor-dependent reductions in mAHP amplitude involve receptors outside the territory of C-bouton synapses. SK channel clusters preferentially appose large cholinergic C-bouton terminals (*Deardorff, 2013*). However, SK channels are also distributed throughout laminae VII and IX, and could therefore be present on more distal motoneuron dendrites where they might be subject to modulation by as yet uncharacterized, Pitx2-negative, populations of cholinergic spinal interneurons (*Bertrand and Cazalets, 2011*; *Gillberg et al., 1988*; *Gillberg et al., 1990*; *Sherriff and Henderson, 1994*; *Deardorff, 2013*). Another possible explanation for the mechanistic differences reported in the present study is that there may be differential, state-dependent levels of involvement of specific components of the C-bouton synapse, depending on the activity levels of motoneurons and V0c interneurons (*Deardorff et al., 2014*). For example, Kv2.1 channel-mediated modulation might dominate at the level of V0c activity induced by our DREADD-based stimulation, but calcium-dependent potassium channels might contribute at different rates of V0c interneuron firing. Testing this hypothesis will require the development of methods that enable finer control of V0c interneuron activity than were possible using the current DREADD-based approach.

Interestingly, in the current study, we found that activation of V0c interneurons was associated with a small increase in both the mAHP measured after single spikes, and the inter-spike AHP measured during repetitive firing. Given that both effects were blocked by guangxitoxin-1E, we propose that this reflects greater recruitment of Kv2.1 channels. This mechanism contrasts previous reports of a substantial reduction in the mAHP due to modulation of calcium-dependent potassium channels (*Miles et al., 2007*). However, as discussed above, these previous data were obtained using less direct, global activation of M2 receptors and reflected measurements of mAHPs evoked by single action potentials only. Furthermore, it remains to be determined whether these effects reflect

activation of M2 receptors at C-bouton synapses. If these effects are C bouton-mediated, perhaps under certain conditions large changes in the mAHP, due to SK channel modulation, reduce the refractory period to allow greater firing frequencies. Meanwhile, in the absence of such large reductions in the mAHP, smaller Kv2.1-mediated increases in the amplitude of the inter-spike AHP may facilitate the removal of inactivation of sodium channels, enabling sustained, higher frequency firing in response to depolarizing stimuli, as has been shown in other neuron types (*Liu and Bean, 2014*; *Johnston et al., 2008*). This Kv2.1 channel-dependent mechanism is also supported by recent work in rat lumbar motoneurons, which has demonstrated that Kv2 channels permit greater motoneuron firing frequencies by maintaining hyperpolarized inter-spike membrane potentials (*Romer et al., 2019*). Another, likely linked, observation in our study was a Kv2.1 channel-dependent narrowing of action potentials following C-bouton activation. This is again consistent with recent reports of increased action potential half-width and a slowing of motoneuron firing when Kv2 channels are blocked (*Romer et al., 2019*; *Fletcher et al., 2017*).

While our findings are consistent with previous work demonstrating that Kv2.1 channels help maintain appropriate levels of repetitive firing in motoneurons (*Romer et al., 2019*; *Fletcher et al., 2017*), the intracellular mechanisms by which M2 receptor activation at C bouton synapses leads to changes in Kv2.1 channel function and motoneuron firing remain unclear. One potential pathway involves the calcineurin-meditated dephosphorylation of Kv2.1 channels. It has been proposed that inhibition of N-type $Ca^{2+}$ channels, known to be located at C-bouton synapses (*Wilson et al., 2004*), by $G_{i/o}$-dependent pathways following M2 receptor activation could lower the levels of local intracellular $Ca^{2+}$, thus reducing calcineurin-related dephosphorylation of Kv2.1 channels (*Deardorff et al., 2014*; *Romer et al., 2019*). Kv2.1 dephosphorylation in response to prolonged excitatory drive is thought to cause channel delustering in motoneurons, whilst reduced activity can increase Kv2.1 channel clustering (*Romer et al., 2019*; *Romer et al., 2016*; *Romer et al., 2014*). Interestingly, in motoneurons, declustering occurs in response to increased glutamatergic, but not cholinergic, input (*Romer et al., 2019*). Thus, activation of C-boutons should not lead to declustering of Kv2.1 channels and has in fact been proposed to maintain Kv2.1 clusters and promote repetitive firing at physiological activity levels (*Romer et al., 2019*). In addition to effecting channel clustering, the phosphorylation state of Kv2.1 channels has been shown to influence their gating parameters. Although data are lacking for motoneurons, dephosphorylation of Kv2.1 channels leads to a hyperpolarizing shift in channel activation and inactivation properties in hippocampal neurons, which is thought to suppress action potential firing frequency (*Mohapatra and Trimmer, 2006*; *Mohapatra et al., 2009*; *Misonou et al., 2005*; *Park et al., 2006*). Thus, it seems plausible that C-bouton activation helps maintain the phosphorylation state of Kv2.1 channels required to support motoneuron firing. It will therefore be interesting to directly test this, and other possible intracellular mechanisms, in future work. Regardless of the exact mechanisms, C-boutons appear well placed to ensure that motoneuron excitability and output is maintained or increased in a task or state-dependent manner, even in the face of elevated glutamatergic drive that would otherwise reduce motoneuron output via delustering of Kv2.1 channels.

An important consideration when interpreting our results is that our experiments were performed on neonatal tissue, because whole spinal cord preparations are not viable when obtained from older animals. Although our in vitro approach allows us to access, isolate and interrogate specific pathways, which are unlikely to be accessible or separable using in vivo approaches, an important caveat is the possibility of developmental changes in C-bouton signaling. The overall pre- and post-synaptic components of C-bouton synapses, including M2 receptors and Kv2.1 channels, appear to remain consistent from early neonatal stages into adulthood (*Zagoraiou et al., 2009*; *Wilson et al., 2004*). Rather than changes in the major components, postnatal maturation of C-boutons involves increased clustering of Kv2.1 channels and M2 receptors along with an increase in the size of C-boutons until motor networks become functionally mature (~P15) (*Wilson et al., 2004*; *Phelps et al., 1984*; *Wetts and Vaughn, 2001*). Given that maturation of spinal networks is sometimes accompanied by changes in the effects of modulators (e.g. 5-HT and acetylcholine [*Jordan et al., 2014*; *Schmidt and Jordan, 2000*]), we cannot exclude the possibility that the exact mechanisms of action at fully mature C-boutons differs from those revealed at neonatal stages. However, the fact that genetic perturbation of C-boutons in adult mice impairs the ability of motoneurons to increase muscle activation in response to behavioral demands (*Zagoraiou et al., 2009*), supports a consistency in the cellular effects of the C-bouton system into adulthood. Recordings from adult motoneurons within in

vivo preparations, and associated modelling studies, have shown state-dependent changes in motoneuron properties including a decrease in AHP amplitude, hyperpolarization of action potential threshold, and a small increase in action potential half-width (*Power et al., 2010*; *MacDonell et al., 2015*; *Krawitz et al., 2001*; *Brownstone et al., 1992*; *Dai et al., 2002*). However, it is difficult to reconcile these results with those of the current study, since the in vivo findings reflect the net effect of multiple parallel modulatory processes and focusses on differences between quiescent states and baseline activity during locomotion or scratching, behavioral states during which the C-bouton system is unlikely to be strongly engaged (*Zagoraiou et al., 2009*). It will therefore be important in future studies to establish methods for interrogating the specific actions and mechanisms of C-bouton-mediated modulation of adult motoneurons during motor tasks of varying intensities.

Cholinergic pathways within the nervous system process a variety of inputs which enable them to fulfil roles in attentional modulation, enhancing responses to stimuli in a state-dependent manner (*Ye et al., 2009*; *Mena-Segovia, 2016*; *Wikman et al., 2019*; *Herrero et al., 2008*). The premotor, intraspinal cholinergic system, involving V0c interneurons and C-boutons, appears to parallel these supraspinal cholinergic systems. V0c interneurons receive synaptic inputs from a range of sources, including sensory pathways, supraspinal structures and CPG networks (*Figure 9*; *Zagoraiou et al., 2009*; *Stepien et al., 2010*; *Murray et al., 2018*), which 'mirror' inputs received by motoneurons and are therefore relevant to desired motor output. The C-bouton system seems to be less active during regular motor tasks such as walking but is particularly important during locomotor behaviors that require increased muscle activation such as swimming (*Zagoraiou et al., 2009*). Thus, V0c interneurons and their C-boutons are thought to provide task-dependent modulation of motor output (*Zagoraiou et al., 2009*). Here, we report novel findings regarding both the organization and mechanisms of action of intraspinal cholinergic modulatory systems. We provide evidence in support of a modular architecture for spinal cholinergic systems, in line with that recently proposed for the locus coerleus (*Chandler et al., 2019*; *Totah et al., 2018*), where V0c interneurons represent one functionally distinct neuromodulatory module devoted to task-dependent control of the intensity of motor output. Meanwhile, we provide the first demonstration of the synaptic and cellular mode of action of C-bouton inputs to motoneurons, demonstrating a role in shaping the action potential and therefore firing patterns via modulation of Kv2.1 channels. Given similarities in the roles of supraspinal and spinal cholinergic modulatory systems in task or state-dependent modulation, the insight gained from our study is likely to provide new insight into cholinergic modulatory systems throughout the mammalian nervous system.

## Materials and methods

### Animal ethics

All the procedures performed on animals were conducted in accordance with the UK Animals (Scientific Procedures) Act 1986 and were approved by the University of St Andrews Animal Welfare Ethics Committee. Experiments on animals performed in the Biomedical Research Foundation of the Academy of Athens were approved by the competent veterinary service of the Prefecture of Athens, Greece in accordance with the existing legal framework. The facility is registered as a 'breeding' and 'user' establishment by the Veterinary Service of the Prefecture of Athens according to the Presidential Decree 56/2013 in harmonization with the European Directive 2010/63/EU for the protection of animals used for scientific purposes.

### Tissue preparation for in vitro electrophysiology and $Ca^{2+}$ imaging

Whole or hemisected spinal cord preparations and spinal cord slices were obtained from postnatal day (P)2-P7 mice. Spinal cords were isolated as previously described *Jiang et al. (1999)*. In brief, animals were euthanized using cervical dislocation and quickly decapitated and eviscerated. They were then pinned ventral side up in a chamber filled with 'dissecting' artificial cerebrospinal fluid (aCSF) continuously gassed with 95% $O_2$ and 5% $CO_2$ at a temperature of ~4°C. The spinal vertebrae were cut and the spinal cord isolated from mid-cervical to upper sacral segments.

To perform ventral root recordings, dorsal roots were trimmed and ventral roots from L1-L5 were kept intact. For patch-clamp recordings in whole spinal cords, access to motoneurons was facilitated by making small vertical cuts in the meninges on the ventral surface. To produce hemisected spinal

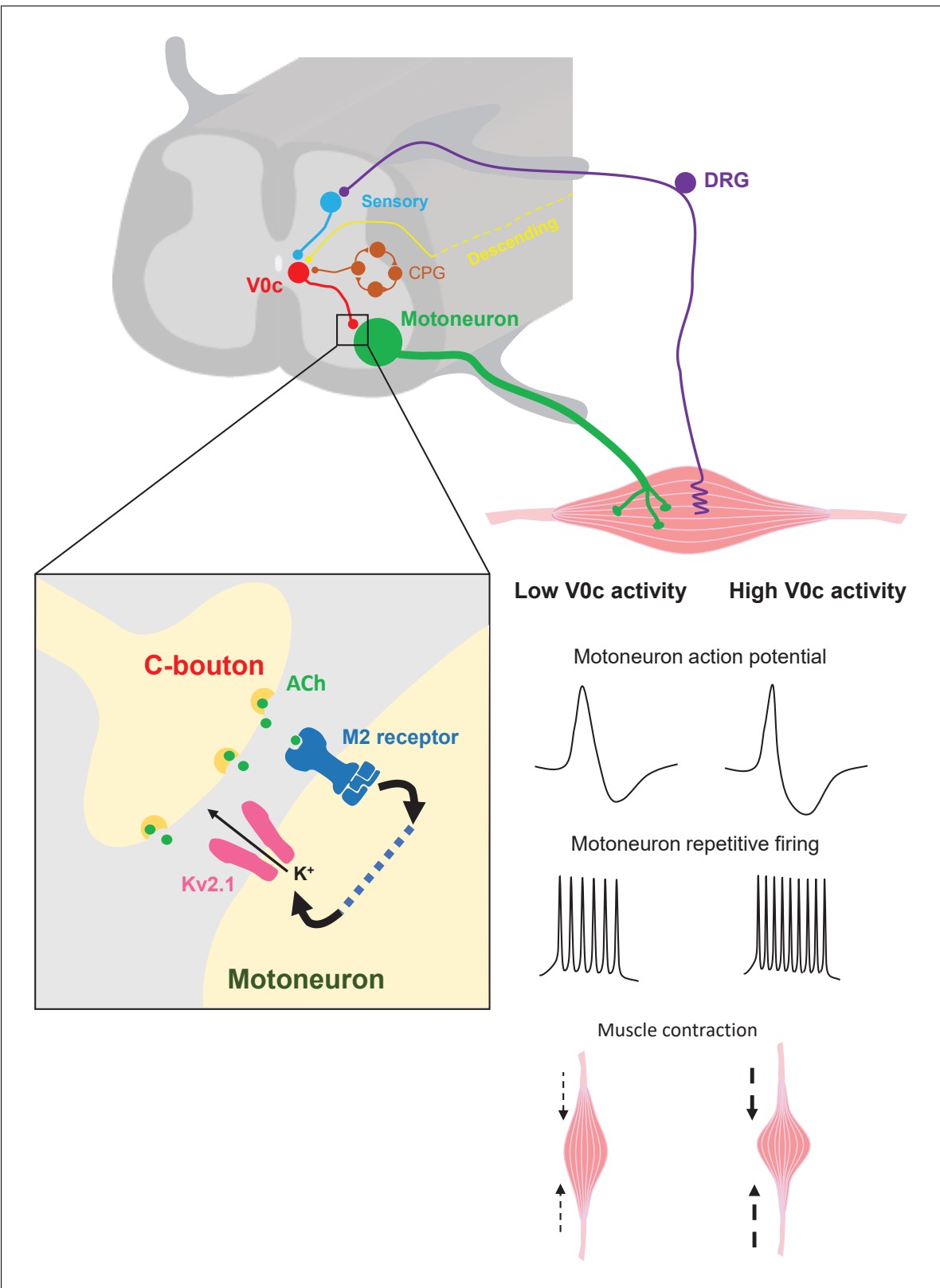

**Figure 9.** Mechanisms of V0c interneuron-mediated modulation of motor output. V0c interneurons (red) form C-bouton synapses on motoneurons (green) and are known to received disynaptic inputs from sensory afferents (blue) as well as input from local CPG neurons (orange) and higher brain regions (yellow) (*Zagoraiou et al., 2009*). Activation of V0c interneurons will lead to release of acetylcholine at C-bouton synapses, activation of postsynaptic M2 muscarinic receptors on motoneurons, and subsequent facilitatory regulation of Kv2.1 channels via as yet uncharacterized signaling

*Figure 9 continued on next page*

Figure 9 continued

pathways (dotted blue line). This modulation leads to an increase in inter-spike AHP and a decrease in action potential duration, allowing for increased and sustained motoneuron firing output, which will translate into more intense muscle contraction (DRG = dorsal root ganglion).

cords, the pia was carefully removed and the two halves cleaved gradually with a dissecting pin. To obtain spinal cord slices, both the ventral and dorsal roots were trimmed and transverse slices were prepared from lumbar segments (300 μm thick) using a vibrating microtome (Leica VT1200). Slices were transferred to a 'recovery' aCSF solution continuously gassed with 95% $O_2$ and 5% $CO_2$ and kept at ~34°C for 45–60 min. Slices were then transferred to a beaker with 'recording' aCSF gassed with 95% $O_2$ and 5% $CO_2$ at room temperature (~20°C).

## Sections for immunofluorescence

Spinal cords were isolated from P2, P7 and P25 mice. P7 and P25 mice were first perfused with 4% PFA. Spinal cords were then post-fixed in 4% PFA and cryoprotected in 30% sucrose solution. They were embedded in Optical Cutting Temperature compound and 14 μm sections were taken using a Leica cryostat. Immunofluorescence was preformed using the following antibodies: rabbit anti-Pitx2 (1:8K CU1533 Jessel lab), goat anti-ChAT (1:100 AB144P Millipore), rabbit anti-dsRed (1:500 Chemicon), rabbit anti-vAChT (1:8K CU1475 Jessell lab) and ginea pig anti-vAChT (1:12K Fitzerald).

## Animal lines

*Pitx2tm4(cre)Jfm* (*Pitx2::Cre*) *mice* (**Liu, 2003**) were crossed with *Gt(ROSA)26Sortm96.1(CAG-GCaMP6s)Hze*/J (*GCAMP6*, Jackson Laboratories, USA; stock #024106) animals to obtain *Pitx2::Cre;GCAMP6* mice that express the GCAMP6 $Ca^{2+}$ indicator in Pitx2+ interneurons (**Acton et al., 2018**). *Pitx2::Cre* mice were mated with homozygous *Gt(ROSA)26Sortm14(CAG-tdTomato)Hze*/J (*tdTomato*) fluorescent reporter animals (**Madisen et al., 2010**), generating *Pitx2::Cre;tdTomato* mice, which allow fluorescent Pitx2+ interneurons to be targeted for single-cell electrophysiology (**Zagoraiou et al., 2009**). *Pitx2::Cre* and *Pitx2::Cre;tdTomato* mice were crossed with *Gt(ROSA)26Sortm2(CAG-hM3Dq *,-mCitrine)Ute*/J animals (*hM3Dq*; Jackson Laboratories, USA; stock #026220) in order to generate *Pitx2::Cre;hM3Dq* and *Pitx2::Cre;tdTomato;hM3Dq* offspring respectively, which express the excitatory hM3Dq DREADD receptor in Pitx2+ interneurons. *Pitx2::Cre* and *Pitx2::Cre;tdTomato* mice were also crossed with *Gt(ROSA)26Sortm1(CAG-hM4Di*,-mCitrine)Ute*/J (*hM4Di*; Jackson Laboratories, USA; stock #026219) to obtain *Pitx2::Cre;hM4Di* and *Pitx2::Cre;tdTomato;hM4Di* animals respectively that express the inhibitory hM4Di DREADD receptor in Pitx2+ interneurons.

The *vAChT-loxP-STOP-loxP-DTA* targeting vector was constructed by inserting a *loxP-PGKneo-triple polyA-loxP-DTA-polyA* cassette at the ATG of the *vAChT* coding sequence. The *vAChT-loxP-STOP-loxP-DTA* targeting vector was electroporated into mouse embryonic stem (ES) cells (129sv/ev), selected with G418 and homologous recombinants were identified by Southern blot analysis. Targeted ES cells were then microinjected into blastocysts and chimeras were crossed to C57BL/6J females resulting the *vAChT-stop-DTA* mouse line. *Pitx2::Cre* and *Pitx2::Cre;tdTomato* mice were crossed with *vAChT-stop-DTA* animals to produce *Pitx2::Cre;vAChT-stop-DTA* and *Pitx2::Cre;tdTomato;vAChT-stop-DTA* offspring respectively in which cholinergic Pitx2+ interneurons (V0c) were selectively ablated due to expression of diphtheria toxin A. Wild Type C57/BL6 mice were also used for some ventral root recordings. All these animals were bred to a C57/BL6 background.

## In vitro $Ca^{2+}$ imaging

Imaging of hemisected spinal cords was performed at 20–24°C. Image acquisition was controlled with Andor Solis (Andor, Oxford Instruments) software. Images were acquired with a Zyla 4.2 scientific CMOS camera using a x40 water immersion objective lens (0.9 numerical aperture). Images were acquired with a rolling shutter at 5 Hz with a 50 ms exposure time. Illumination was provided by a 470 nm CoolLED system. Fictive locomotor output was recorded from L2/3 ventral roots during image acquisition (see below). Imaging and ventral root recordings were synchronized using a TTL pulse from the Digidata 1440 A/D to the CMOS camera.

## In vitro electrophysiology

Spinal cord slices or intact spinal cord preparations were immersed in a recording chamber perfused with recording aCSF (approximately 1 mL per second). Whole-cell patch-clamp recordings were established using borosilicated glass microelectrodes (2.5–6 MΩ) filled with intracellular solution. Signals were amplified and filtered (4 kHz low-pass Bessel filter) with a MultiClamp 700B amplifier (Molecular Devices, Sunnyvale, CA) and acquired at ≥10 kHz using a Digidata 1440A A/D board and pClamp software (version 10.6, Molecular Devices, Sunnyvale, CA).

Measurements of drug-induced currents and input resistance (using 2.5 mV steps from −75 to −52.5 mV) were conducted in voltage-clamp mode ($V_{hold}$, −60 mV). Firing output was measured in current-clamp mode using either 'gap-free' acquisition for spontaneous firing, by injecting a series of 1 s square current pulses (from 10 pA, 50 pA increments) to induce repetitive firing, by applying 10 ms supramaximal pulses to evoke single action potentials, or via injection of a single supramaximal current ramp (1 s duration) to investigate firing threshold. In order to facilitate comparisons, a bias current was applied as necessary during current-clamp protocols to maintain the resting potential of neurons (∼−60 mV).

Ventral root recordings were performed in whole or hemisected (in parallel with $Ca^{2+}$ imaging) spinal cords by attaching suction electrodes to the ventral roots (L1-L5). Rhythmic locomotor-related activity was induced by applying 5-hydroxytryptamine (5-HT; 10 μM), N-methyl-D-aspartate (NMDA; 5 μM) and dopamine (DA, 50 μM). Signals were amplified and filtered (band-pass filter 30–3000 Hz, Qjin Design) and then acquired at a frequency of 6 kHz with a Digidata 1440A A/D board and pClamp software. Custom built amplifiers (Qjin design) allowed acquisition of raw signals with simultaneous online rectification and integration (50 ms time constant).

## Drugs and solutions

The dissecting aCSF contained (in mM): 25 NaCl, 188 sucrose, 1.9 KCl, 1.2 $NaH_2PO_4$, 10 $MgSO_4$, 1 $CaCl_2$, 26 $NaHCO_3$, 25 glucose and 1.5 kynurenic acid. The recovery solution contained (in mM): 119 NaCl, 1.9 KCl, 1.2 $NaH_2PO_4$, 10 $MgSO_4$, 1 $CaCl_2$, 26 NaHCO3, 20 glucose and 1.5 kynurenic acid. The recording aCSF contained (in mM): 127 NaCl, 3 KCl, 1.25 $NaH_2PO_4$, 1 $MgC_{l2}$, 2 $CaCl_2$, 26 $NaHCO_3$, 10 glucose. The intracellular solution for patch-clamp recordings contained (in mM): 140 $KMeSO_4$, 10 NaCl, 1 $CaCl_2$, 10 HEPES, 1 EGTA, 3 Mg-ATP and 0.4 GTP-$Na_2$ (pH 7.2–7.3, adjusted with KOH). NMDA, DA, 5-HT and methoctramine were supplied by Sigma-Aldrich; Clozapine-N-oxide (CNO) by Tocris and Hello-Bio; and guangxitoxin-1E by Alomone Labs. All drugs were dissolved in $H_2O$.

## Data analysis

Analysis of $Ca^{2+}$ imaging was performed with FIJI software. Data were first converted to 8 bit and processed with a background subtraction and 3D smoothing. If required, a histogram-based bleach correction was performed, along with manual drift correction to control for tissue movement. Active cells were selected and delineated from the images, guided by a maximum intensity projected image to highlight high-intensity structures. Intensity measurements were converted to Δf/f0, calculated as: *100 x (fluorescence value – baseline fluorescence ÷ baseline fluorescence)*. Baseline fluorescence was calculated as the mean intensity from 10 frames during a period of low-level activity for each cell within the first minute of recording. $Ca^{2+}$ transients from Pitx2$^+$ interneurons in hemisected spinal cords were detected and quantified using DataView software (courtesy of Dr W. J. Heitler, University of St Andrews). Circular phase plots were used to depict the coupling of cellular $Ca^{2+}$ activity with ventral root bursts during fictive locomotion. The beginning of the locomotor cycle, marked as 0 at the top of circular plots, was defined as the onset of flexor-related bursts recorded from upper lumbar ventral roots. Rayleigh's test for uniformity was used to statistically assess whether cells were significantly coupled to bursts of locomotor related activity.

In single-cell voltage clamp recordings, changes in holding current were calculated as the difference between the current value immediately before drug perfusion and the maximum change in current induced by the drug. Voltage-current relationships were used to calculate input resistance before and after drug perfusion. In current clamp experiments, rheobase was measured as the first current step eliciting firing. The current required to generate a depolarizing block was measured from the first step in which motoneurons stopped firing repetitively during the 1s-long pulse. This

was defined as a $\geq$ 100 ms silent period following the establishment of repetitive firing during current injection. Maximum firing frequencies were measured from the current step eliciting the highest firing rate. The magnitude of the inter-spike afterhyperpolarization was calculated from the last five spikes of the step that elicited the maximum firing rate. For single action potential analysis, mAHP amplitude was calculated as the difference between the resting voltage before injecting the 10ms-long depolarizing current and the peak value of the mAHP (averaged across 15 traces). For these single spikes, a differential function (dV/dt) was calculated and the onset of the action potential was set when dV/dt reached 10 mV/ms, which was then used for the estimation of half-width. The voltage threshold for action potentials was calculated from the first action potential elicited by current ramps.

Ventral root bursts were analyzed offline using DataView software. Bursts were identified from the integrated/rectified trace from which frequency and duration were measured. Burst amplitude was calculated from the respective segment of the raw trace. Data were averaged in 0.5 min time bins and normalized to a 10 min pre-control period to construct time course plots. Statistical comparisons were made on raw data averaged over 5 min periods in each condition.

## Statistical analysis

Data are represented in the text as mean $\pm$ s.e.m. In the figures, values are depicted as box plots displaying the distribution of data as the minimum, first quartile, median, third quartile, and maximum value for each dataset. In patch-clamp and $Ca^{2+}$ imaging experiments each 'n' corresponds to one cell whereas in ventral root recordings it corresponds to one whole spinal cord preparation. D'Agostino-Pearson omnibus tests were used to access normality. Repeated measures ANOVA with Tukey's multiple comparison test (for normal distributed data) or Friedman test with Dunns' post-test (for non-normal distributed data) were used for the analysis of ventral root bursts between control, drug and washout. Rayleigh's test was used to assess the mean phasing of Pitx2$^+$ interneuron activity relative to ventral root bursting. Paired $t$-test or Wilcoxon signed-rank test were used to compare data from single-cell experiments between control and drug. One-way ANOVA with Tukey's post-test was used to compare properties of Pitx2$^+$ interneurons between DREADD and control mice. Unpaired t-test or Mann-Whitney U-test were used to compare data from $Ca^{2+}$ imaging, immunohistochemistry, ventral root duration and frequency, and properties of motoneurons from DREADD and control mice. Values of p<0.05 were considered statistically significant.

## Acknowledgements

F Nascimento was supported by The Alfred Dunhill Links Foundation. G B Miles and M J Broadhead received support from Biotechnology and Biological Sciences Research Council Grant BB/M021793/1. L Zagoraiou and E Tsape were supported by Fondation Santé.

## Additional information

### Funding

| Funder | Grant reference number | Author |
|---|---|---|
| Alfred Dunhill Links Foundation | | Filipe Nascimento |
| Biotechnology and Biological Sciences Research Council | BB/M021793/1 | Matthew James Broadhead Gareth Miles |
| Foundation Santé | | Eirini Tsape Laskaro Zagoraiou |

The funders had no role in study design, data collection and interpretation, or the decision to submit the work for publication.

### Author contributions

Filipe Nascimento, Conceptualization, Data curation, Formal analysis, Validation, Investigation, Visualization, Methodology, Writing - original draft, Project administration, Writing - review and editing;

Matthew James Broadhead, Conceptualization, Data curation, Formal analysis, Validation, Investigation, Visualization, Methodology, Writing - original draft, Writing - review and editing; Efstathia Tetringa, Eirini Tsape, Formal analysis, Validation, Investigation, Methodology; Laskaro Zagoraiou, Conceptualization, Resources, Formal analysis, Supervision, Funding acquisition, Validation, Visualization, Methodology, Writing - original draft, Project administration, Writing - review and editing; Gareth Brian Miles, Conceptualization, Resources, Data curation, Formal analysis, Supervision, Funding acquisition, Validation, Visualization, Methodology, Writing - original draft, Project administration, Writing - review and editing

### Author ORCIDs
Filipe Nascimento (iD) https://orcid.org/0000-0002-9426-2807
Matthew James Broadhead (iD) http://orcid.org/0000-0002-4078-5581
Gareth Brian Miles (iD) https://orcid.org/0000-0002-8624-4625

### Ethics
Animal experimentation: All the procedures performed on animals were conducted in accordance with the UK Animals (Scientific Procedures) Act 1986 and were approved by the University of St Andrews Animal Welfare Ethics Committee. Experiments on animals performed in the Biomedical Research Foundation of the Academy of Athens were approved by the competent veterinary service of the Prefecture of Athens, Greece in accordance with the existing legal framework. The facility is registered as a 'breeding' and 'user' establishment by the Veterinary Service of the Prefecture of Athens according to the Presidential Decree 56/2013 in harmonisation with the European Directive 2010/63/EU for the protection of animals used for scientific purposes.

### Decision letter and Author response
Decision letter https://doi.org/10.7554/eLife.54170.sa1
Author response https://doi.org/10.7554/eLife.54170.sa2

## Additional files

### Supplementary files
• Source data 1. Values for properties of motoneurons, Pitx2$^+$ interneurons and ventral root output in control and DREADD mice before CNO perfusion.

• Supplementary file 1. Single-cell properties of motoneurons and Pitx2$^+$ interneurons and ventral root output are not different between DREADD and control mice prior to the application of CNO.

• Transparent reporting form

### Data availability
All of the data presented in this study are included in the manuscript and supporting files.

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
