## [Decision Letter]

**Acceptance summary:**

Your study provides important new understanding of the mechanisms underlying the modulation of motoneurons by cholinergic Pitx2^+^ interneurons. The additional experiments and explanations provided to the revised manuscript nicely strengthened your claims.

**Decision letter after peer review:**

Thank you for submitting your article "Synaptic mechanisms underlying modulation of locomotor-related motoneuron output by premotor cholinergic interneurons" for consideration by *eLife*. Your article has been reviewed by three peer reviewers, and the evaluation has been overseen by a Reviewing Editor and Ronald Calabrese as the Senior Editor. The following individuals involved in review of your submission have agreed to reveal their identity: Adam Deardorff (Reviewer #2); Larry Jordan (Reviewer #3).

The reviewers have discussed the reviews with one another and the Reviewing Editor has drafted this decision to help you prepare a revised submission.

Summary:

This is a very nice study addressing fundamental questions regarding cholinergic neuromodulation within the mammalian spinal cord and, more specifically, cellular mechanisms underlying the state dependent regulation of motoneurons by C-boutons. The experiments are well designed and build upon prior investigations, several of which were performed in the authors' own laboratories. The data are robust, novel, and represent a significant contribution to the field of motoneuron/movement neuroscience and will be of broader interest to those studying neuromodulation and synaptic signaling.

In particular, this work provides novel information about the role of Pitx2 inter-neurons (the source of C terminals on motoneurons) using calcium imaging, chemogenetics to excite or inhibit these cells specifically, as well as a novel method to ablate V0c interneurons. The authors show that, in contrast to earlier work, the effect of specific manipulation of the Pitx2 neurons involves actions not only at the M2 receptor but also modulates the Kv2.1 channels located at C terminals. Instead of a decrease in mAHP, as shown before and used as a basis for C-terminal increases in motoneuron excitability, this genetic approach revealed increased AHP and narrowed spikes to explain the increased excitability. This is a marked change from the previously accepted basis for the increase in motoneuron excitability. The authors also showed that c-boutons hyperpolarize motoneuron threshold, another factor that could increase motoneuron excitability. They were able to distinguish effects of muscarinic receptors from those involving Kv2.1 using specific blockers.

Please read the list of essential revisions requested by the three reviewers below, address all points raised and resubmit your revised manuscript after completion of text edits and a moderate amount of experiments (mainly precisions and controls).

Essential revisions:

– The authors do not fully explain how these opposite effects on mAHP can lead to the same result.

– The authors were fortunate that the affinity of CNO for 5-HT receptors involved in locomotion did not interfere with fictive locomotion, since the affinity of CNO for 5-HT receptors is roughly the same as its affinity for the DREADD receptors. Perhaps this issue could be addressed.

– Data from adult animals with the DREADD receptors would have been useful. There is evidence that the response to ACh in adults differs from that in neonatal animals, such that in adult spinal animals the effect of a muscarinic antagonist has the opposite effect to that observed in neonates. This issue might be discussed.

– Developmental changes in the effects of C terminal activation is especially relevant because all of the recordings from motoneurons in adult cats and rats (during fictive locomotion or scratch e.g., Brownstone, et al., 1992; MacDonell et al., 2015; Power et al., 2010) show a decrease in the AHP. The data on spike duration is shown in Krawitz et al., 2001, and modeling studies (Dai et al., 2002) showed that action potential shape changes occur with modeled alterations in the delayed rectifier. There seems to be no definitive work in adult animals showing a change in action potential width during fictive locomotion consistent with the current results, these results should be discussed. Changes in the effects of ACh and 5-HT during development have been observed. It is important to show in this study that the observations in the neonatal mouse persist into adulthood, otherwise the data is applicable only to motoneurons control during development. The importance of such a developmental role should be discussed, in the absence of recordings from adult animals to confirm that the observations here are relevant to adults.

– Substantive concerns: CNO may be converted into clozapine, a 5-HT2a antagonist (Thompson et al. ACS Pharmacol Transl Sci. 2018 Sep 14;1(1):61-72. doi: 10.1021/acsptsci.8b00012). Perhaps a few control experiments using C21, a compound that has been suggested to have less side-effects than CNO would be recommended, though the authors do make some control experiments where CNO is non-hM4Di expressing spinal cords are done. This important control is done sometimes but not for all experiments (e.g. recordings of motoneurons to CNO were not done in non-hM4Di spinal cords).

Fictive locomotion recordings show L2 activity. Since calcium imaging of Pitx2 neurons seems to suggest they are flexor related, were the effects observed only constrained to flexor-dominant L2 ventral roots and not extensor-dominant L5 ventral roots?

– The major finding is a prominent role for Kv2.1 in MN excitability, and that Kv2.1 is regulated by C-boutons. These data significantly advance prior studies, which are appropriately cited by the authors. I am concerned about one aspect of the interpretation. The authors state that calcium/calcineurin dependent Kv2.1 channel dephosphorylation results in "reduced K^+^ conductance". However, the cited studies by Trimmer and others show Kv2.1 channel dephosphorylation causes a leftward, hyperpolarizing shift in the voltage dependence of activation, such that the normalized conductance of dephosphorylated Kv2.1 is increased at a given membrane potential. Conversely, normalized conductance is reduced when Kv2.1 is phosphorylated. This should be corrected and the subsequent Discussion section revised to reflect this correction.

– Did the authors analyze the effects of GxTx alone on single action potentials? If not, could the authors comment on the specificity of GxTx for Kv2 vs. Kv4 channels as inhibition of an A current could also broaden AP half width.

– The authors state that C-bouton activation recruits Kv2.1 channels (Discussion, seventh paragraph), and Figure 9 implies a direct link between the m2 receptors and Kv2.1 channels. Some clarification here would be helpful. M2 receptors typically couple to Gi/Go coupled pathways that inhibit calcium channels (β subunit) or decrease cAMP/PKA activity (α subunit). Could the influence of m2 receptors on Kv.2.1 channels be via effects on intracellular calcium or other phosphorylation cascades? Or is there evidence that G protein α or β subunits directly activate Kv2.1 channels (in MNs or other cell types)?

– The result that V0c activity increases rather than decreases MN mAHP amplitude, and that this effect is blocked by methoctramine, is surprising given the author's prior work that local application of muscarine ablates the mAHP via m2 receptors (Miles et al., 2007). The authors should be lauded for publishing data which, on the surface, appears at odds with this highly cited study. Their suggestion that m2 receptor reduction in mAHP occurs via receptors "outside the territory of C-bouton synapses" is plausible and intriguing, but would be strengthened if the authors could briefly address 1) a possible mechanism by which SK channels are activated by more distant m2 receptors or evidence for similar findings in other cell types and 2) the functional utility of C-bouton postsynaptic SK expression if the channels are not involved in m2 mediated C-bouton signaling.

– Related to # above, m2 receptor labelling has been plagued by non-specific commercial antibodies. Though some studies cited by the authors describe a more uniform distribution of m2 receptors across the MN membrane (Welton et al., 1999; Hellstrom et al., 2003), a result suggested physiologically by the authors' prior work (Nascimento et al., 2019), other studies cited show a more clustered distribution of m2 receptors specifically localized to C-bouton postsynaptic sites (Wilson et al., 2004; Meunnich and Fyffe, 2004; Deardorff et al., 2014). The existence of extra synaptic m2 receptors is therefore not as definitive as the authors state (Discussion, fourth paragraph) – but again is certainly plausible.

– Are there changes in MN properties when Pitx2 cells are inhibited?

---

## [Author Response]

Essential revisions:– The authors do not fully explain how these opposite effects on mAHP can lead to the same result.

We thank the reviewers for recognising the importance of reporting these new findings despite the fact that they appear at odds with our previous work (mentioned below). Upon reflection, we realise we could have discussed this more thoroughly. We have therefore expanded our Discussion (eighth paragraph) of these differences, including whether or not previous mechanisms were of C-bouton origin and whether the different mechanisms are truly mutually exclusive or whether their relative influence could depend on neuronal/network states. We have also tried to better highlight potential differences between calcium-dependent mAHP and the Kv2.1 dependent effect on spike repolarisation and inter-spike AHP. In addition, we have tried to stress the likelihood that the most influential effect of Kv2.1 channel modulation may relate to changes in action potential kinetics.

– The authors were fortunate that the affinity of CNO for 5-HT receptors involved in locomotion did not interfere with fictive locomotion, since the affinity of CNO for 5-HT receptors is roughly the same as its affinity for the DREADD receptors. Perhaps this issue could be addressed.We have now addressed this issue further in our Discussion (third paragraph). In addition to the previous data on controls for CNO on whole cord and motoneuron outputs (supplementary figures), we have also added some new additional data on controls for CNO on motoneuron mAHP and subthreshold properties in animals that do not express DREADD receptors (Figure 7—figure supplement 1).– Data from adult animals with the DREADD receptors would have been useful. There is evidence that the response to ACh in adults differs from that in neonatal animals, such that in adult spinal animals the effect of a muscarinic antagonist has the opposite effect to that observed in neonates. This issue might be discussed.– Developmental changes in the effects of C terminal activation is especially relevant because all of the recordings from motoneurons in adult cats and rats (during fictive locomotion or scratch e.g., Brownstone, et al., 1992; MacDonell et al., 2015; Power et al., 2010) show a decrease in the AHP. The data on spike duration is shown in Krawitz et al., 2001, and modeling studies (Dai et al., 2002) showed that action potential shape changes occur with modeled alterations in the delayed rectifier. There seems to be no definitive work in adult animals showing a change in action potential width during fictive locomotion consistent with the current results, these results should be discussed. Changes in the effects of ACh and 5-HT during development have been observed. It is important to show in this study that the observations in the neonatal mouse persist into adulthood, otherwise the data is applicable only to motoneurons control during development. The importance of such a developmental role should be discussed, in the absence of recordings from adult animals to confirm that the observations here are relevant to adults.

We acknowledge that a major caveat of our in vitro approach is that it can only be conducted in neonatal tissue and that the exact mechanisms revealed here may change with development. As mentioned earlier (and now included in our Discussion), differences in the exact pathways engaged may also depend on the specific state of the network and its constituent neurons, which is also not easy to model in vitro. However, we believe that our in vitro approach comes with many important benefits, including providing the ability to more finely control and interrogate cellular mechanisms of action. For example, we have been able to focus on the specific actions of C-bouton activation, which is important due to the revelation that multiple spinal cholinergic pathways are likely to differentially effect motoneuron properties and output.

We have considered whether our experiments could be conducted in older tissue and have therefore consulted Prof Marco Beato (University College London), who routinely performs motoneuron recordings in young adult (P15-P25) mouse spinal cord preparations. Given the intersegmental projection patterns of V0c interneurons, our experiments required recordings in intact spinal cord preparations. However, the advice we have received, alongside our own experience, suggests that it would not be possible to perform these recordings in tissue obtained from young adult mice. This reflects a range of factors including difficulties in visualizing motoneurons because of the greater myelinization at this stage and poor motoneuron survival in older intact preparations where a hypoxic core is produced.

Given that experiments in older preparations are not yet possible, we have added a section to our Discussion in which we consider the developmental changes of C-boutons and spinal motor networks (tenth paragraph). Within this section we also highlight the fact that the synaptic structures we are studying are detectable at birth, and that they then undergo a maturation process, which mostly involves increased receptor/channel clustering, rather than any changes in the complement of proteins at the synapse. We believe this adds support to the notion that mechanisms of C-bouton modulation continue to involve M2 receptors and Kv2.1 channels in adulthood. In addition, we highlight that previous experiments in which C-boutons were genetically perturbed in adult mice support a consistent role for C-boutons in regulating the intensity of motoneuron output.

– Substantive concerns: CNO may be converted into clozapine, a 5-HT2a antagonist (Thompson et al. ACS Pharmacol Transl Sci. 2018 Sep 14;1(1):61-72. doi: 10.1021/acsptsci.8b00012). Perhaps a few control experiments using C21, a compound that has been suggested to have less side-effects than CNO would be recommended, though the authors do make some control experiments where CNO is non-hM4Di expressing spinal cords are done. This important control is done sometimes but not for all experiments (e.g. recordings of motoneurons to CNO were not done in non-hM4Di spinal cords).

Although we agree that this is an important consideration, conversion of CNO to clozapine is only thought to occur in vivo, since hepatic enzymes are required for effective metabolization (MacLaren et al., 2016; Pirmohamed et al., 1995). We believe it is therefore unlikely that off-target effects of clozapine would affect our use of CNO within isolated spinal cord preparations. As mentioned above, we have now addressed this, and related issues, in our revised Discussion (third paragraph).

CNO has also been shown to significantly affect 5-HT2a receptors (~50% inhibition of binding) at a concentration of 10µM (Gomez et al., 2017). In our study we perfused CNO at a lower concentration (1µM). If CNO or CNO-related metabolites had off-target effects, such as on 5-HT2a receptors, we would expect changes in whole network or motoneuron output. However, we have included data showing that, in animals that do not express DREADD receptors, CNO (1µM) had no effect on locomotor-related ventral root output, single cell output, or subthreshold motoneuron properties (Figure 3—figure supplement 1 and Figure 7—figure supplement 1). We have also now added additional experimental data showing a lack of effect of CNO on the mAHP or input resistance of motoneurons in control mice (Figure 7—figure supplement 1).

Although we considered alternatives to CNO, a recent study has indicated that Cmp-21, despite being more selective than CNO in activating DREADD receptors (Thompson et al. ACS Pharmacol Transl Sci. 2018 doi: 10.1021/acsptsci.8b00012), also has off-target effects. This includes actions on 5-HT receptors and muscarinic receptors (including M2 receptors focused on in this study; Jendryka et al. 2019, Sci Rep, https://doi.org/10.1038/s41598-019-41088-2).

Given the potential issues with alternatives and the fact that CNO applied at 1µM failed to elicit any detectable effects at single cell or whole network levels in non-GM in vitro preparations, we do not believe the potential benefits of repeating our experiments with a different agonist outweigh the animal costs.

Fictive locomotion recordings show L2 activity. Since calcium imaging of Pitx2 neurons seems to suggest they are flexor related, were the effects observed only constrained to flexor-dominant L2 ventral roots and not extensor-dominant L5 ventral roots?

In the Ca2+ imaging experiments we imaged all Pitx2+ interneurons (V0c and V0g) in the lumbar region while recording from L1-L3 ventral roots. In our discussion, we hypothesise that their bias in phasing to flexorrelated upper lumbar roots may be attributable to the differing density of V0c and V0g interneurons in the lumbar spinal, which is in agreement with previous reports (Zagoraiou et al., 2009 Neuron). Ventral root recordings performed as parts of DREADD-based and genetic ablation experiments included L1-L5 roots, since both upper and lower lumbar motoneurons are highly innervated by C-boutons. We have now emphasized that we have measured VR output from L1-L3 in the calcium imaging experiments (subsection “Pitx2+ INs are rhythmically active during fictive locomotion”) and L1-L5 in the pharmacological (subsection “Chemogenetic inhibition of Pitx2+ INs decreases the amplitude of locomotor output lines”and subsection “Ablation of V0c INs eliminates muscarinic modulation of the intensity of locomotor output”). Although we believe that C-bouton activity affects the output recorded from all lumbar roots, the possibility of differential effects on different ventral roots was not directly assessed in our experiments. Given that our experiments were not designed to focus on this question, we predominantly recorded from upper lumbar roots, which typically provide the most reliable output. We therefore do not have sufficient data to make comparisons between specific roots and would need a large number of new experiments to achieve this.

– The major finding is a prominent role for Kv2.1 in MN excitability, and that Kv2.1 is regulated by C-boutons. These data significantly advance prior studies, which are appropriately cited by the authors. I am concerned about one aspect of the interpretation. The authors state that calcium/calcineurin dependent Kv2.1 channel dephosphorylation results in "reduced K^+^ conductance". However, the cited studies by Trimmer and others show Kv2.1 channel dephosphorylation causes a leftward, hyperpolarizing shift in the voltage dependence of activation, such that the normalized conductance of dephosphorylated Kv2.1 is increased at a given membrane potential. Conversely, normalized conductance is reduced when Kv2.1 is phosphorylated. This should be corrected and the subsequent Discussion section revised to reflect this correction.

We thank the reviewers for pointing out this error, which we have now corrected in the revised manuscript. We have also rewritten this section of the Discussion and highlighted that future work will need to be done to decipher the exact intracellular pathways involved (Discussion, ninth paragraph).

– Did the authors analyze the effects of GxTx alone on single action potentials? If not, could the authors comment on the specificity of GxTx for Kv2 vs. Kv4 channels as inhibition of an A current could also broaden AP half width.

We did not analyse the effects of GxTX-1E on motoneuron action potentials, although this has been done previously by Fletcher and colleagues (Fletcher et al., 2017). GxTX-1E is reported to be at least 8-times weaker on Kv4 versus Kv2 channels (Herrington et al., 2006). Previous data indicate that it effectively blocks Kv2 at 100nM but Kv4 at higher concentrations (Herrington et al., 2006). We therefore perfused GxTX-1E at 50nM which we believe will only mildly impact Kv4. Previous work has used 100nM of GxTX-1E to selectively block Kv2 channels in both the neonatal mouse spinal cord (Fletcher et al., 2017) and hippocampus (Liu and Bean, 2014). Liu and Bean, 2014, concluded that “…100 nM GxTX-1E as an optimal concentration to use in current-clamp experiments, with nearly saturating effects on Kv2 but minimal effects on IA…”. In addition, Fletcher et al., 2017, reported a lack of Kv4 expression on spinal motoneurons of neonatal mice. We have now included a statement regarding the specificity of GxTX-1E and the concentration used in our revised Results section (subsection “Chemogenetic excitation of Pitx2^+^ INs reveals synaptic mechanisms at C-boutons”, second paragraph).

– The authors state that C-bouton activation recruits Kv2.1 channels (Discussion, seventh paragraph), and Figure 9 implies a direct link between the m2 receptors and Kv2.1 channels. Some clarification here would be helpful. M2 receptors typically couple to Gi/Go coupled pathways that inhibit calcium channels (β subunit) or decrease cAMP/PKA activity (α subunit). Could the influence of m2 receptors on Kv.2.1 channels be via effects on intracellular calcium or other phosphorylation cascades? Or is there evidence that G protein α or β subunits directly activate Kv2.1 channels (in MNs or other cell types)?

As outlined above in our response to a similar query, we have now amended our discussion of the possible pathways by which M2 receptor activation might influence Kv2.1 channels (including those involving intracellular Ca^2+^ and phosphorylation). Given that we are unaware of any clear evidence of direct activation of Kv2.1 channels by M2 receptors, we have also amended Figure 9 and the respective legend, to better represent the unknown nature of this interaction and the likely involvement of intracellular signalling pathways.

– The result that V0c activity increases rather than decreases MN mAHP amplitude, and that this effect is blocked by methoctramine, is surprising given the author's prior work that local application of muscarine ablates the mAHP via m2 receptors (Miles et al., 2007). The authors should be lauded for publishing data which, on the surface, appears at odds with this highly cited study. Their suggestion that m2 receptor reduction in mAHP occurs via receptors "outside the territory of C-bouton synapses" is plausible and intriguing, but would be strengthened if the authors could briefly address 1) a possible mechanism by which SK channels are activated by more distant m2 receptors or evidence for similar findings in other cell types and 2) the functional utility of C-bouton postsynaptic SK expression if the channels are not involved in m2 mediated C-bouton signaling.We thank the reviewers for appreciating our reporting of mechanism that may appear at odds with our previous work. We have now discussed the possibility of SK channels being activated by M2 receptors outside of C-boutons synapses (Discussion, seventh paragraph). We have also expanded our discussion of a possible role for SK channels at C-boutons in the regulation of spike threshold (Discussion, sixth paragraph).– Related to # above, m2 receptor labelling has been plagued by non-specific commercial antibodies. Though some studies cited by the authors describe a more uniform distribution of m2 receptors across the MN membrane (Welton et al., 1999; Hellstrom et al., 2003), a result suggested physiologically by the authors' prior work (Nascimento et al., 2019), other studies cited show a more clustered distribution of m2 receptors specifically localized to C-bouton postsynaptic sites (Wilson et al., 2004; Meunnich and Fyffe, 2004; Deardorff et al., 2014). The existence of extra synaptic m2 receptors is therefore not as definitive as the authors state (Discussion, fourth paragraph) – but again is certainly plausible.

We agree that this is speculative but thank the reviewer for seeing it as plausible. We also believe it is possible that some M2 receptors are expressed at cholinergic synapses on MNs that are not C-boutons, and that they may therefore not technically be extra synaptic.

– Are there changes in MN properties when Pitx2 cells are inhibited?

We have now added a new set of data on DREADD-mediated inhibition of Pitx2^+^ interneurons during fictive locomotion which shows changes in transient current and input resistance of motoneurons (subsection “Chemogenetic inhibition of Pitx2^+^ INs decreases the amplitude of locomotor output”, Figure 4—source data 1).